# Gateways to Tractability for Satisfiability in Pearl's Causal Hierarchy

**Robert Ganian** [1]   **Marlene Gründel** [1]   **Simon Wietheger** [1]

## Abstract

Pearl's Causal Hierarchy (PCH) is a central framework for reasoning about probabilistic, interventional, and counterfactual statements, yet the satisfiability problem for PCH formulas is computationally intractable in almost all classical settings. We revisit this challenge through the lens of parameterized complexity and identify the first gateways to tractability. Our results include fixed-parameter and XP-algorithms for satisfiability in key probabilistic and counterfactual fragments, using parameters such as primal treewidth and the number of variables, together with matching hardness results that map the limits of tractability. Technically, we depart from the dynamic programming paradigm typically employed for treewidth-based algorithms and instead exploit structural characterizations of well-formed causal models, providing a new algorithmic toolkit for causal reasoning.

## 1. Introduction

Pearl's Causal Hierarchy (PCH) (Shpitser & Pearl, 2008; Pearl, 2009; Bareinboim et al., 2022) is a central pillar of the modern theory of causality that is employed in artificial intelligence and other reasoning tasks—see, e.g., the recent book (Pearl, 2018) on the topic. The PCH is a framework that has three basic layers of depth which capture three fundamental degrees of sophistication for analyzing causal effects and relationships. All of these layers provide a means of formalizing statements via formulas capturing the behavior of a set of probabilistic variables in a *Structural Causal Model* (SCM) (Glymour et al., 1987; Pearl, 2009;

Koller & Friedman, 2009; Elwert, 2013), which is a well-established representation of systems with observed as well as hidden variables over a specified domain and their mutual dependencies. As a basic illustrative example, the statement "the likelihood of having both diabetes ($D =$ yes) and blood type B+ ($T =$ B+) is at most 1%" can be expressed by the formula $\psi = \Pr(D = \text{yes} \wedge T = \text{B+}) \leq 0.01$.

The formula $\psi$ above belongs to the first, basic layer of the PCH—that is, the layer $\mathcal{L}_{\text{prob}}$ of probabilistic reasoning that captures direct statements one can make about the probabilities of certain outcomes. The second layer, $\mathcal{L}_{\text{causal}}$, expands on the basic probability terms in $\mathcal{L}_{\text{prob}}$ via the introduction of Pearl's do-operator (Pearl, 2009) which captures interventional causal reasoning. A basic example of an event that can be captured on this layer of the PCH is contracting a disease after being vaccinated against that disease; the probability of this event can be expressed using the term $\Pr([Y = \text{vaccinated}] \, X = \text{contracted})$[1], where the square brackets denote an intervention that is applied before observing the outcome.[2] Hence, the second layer of the PCH allows us to make statements such as $\Pr([Y = \text{vaccinated}] \, X = \text{contracted}) < \Pr(X = \text{contracted})$. The third layer $\mathcal{L}_{\text{counterfact}}$ of the PCH expands on $\mathcal{L}_{\text{causal}}$ by allowing interventions to be chained, and enables complex statements related to counterfactual situations. For instance, a third-layer term such as $\Pr\big(([M = \text{yes}] \, H = \text{no}) \mid (M = \text{no} \wedge H = \text{yes})\big)$ can express the probability that a patient who did not take medication ($M$) and was hospitalized ($H$) would have avoided hospitalization if he had taken the medication. Formal definitions of these as well as related notions are available in Section 2.

While the three layers of depth of the PCH focus on the expressivity inside the probability term $\Pr(\cdot)$, there is a second

*Equal contribution   [1]Algorithms and Complexity Group, TU Wien, Vienna, Austria. Correspondence to: Robert Ganian <rganian@gmail.com>, Marlene Gründel <mgruendel@ac.tuwien.ac.at>, Simon Wietheger <swietheger@ac.tuwien.ac.at>.

*Proceedings of the 43rd International Conference on Machine Learning*, Seoul, South Korea. PMLR 306, 2026. Copyright 2026 by the author(s).

Additional details and full proofs are available on arxiv: https://arxiv.org/abs/2511.08091.

---

[1]Equivalently, $\Pr(X = \text{contracted} \mid do(Y = \text{vaccinated}))$. We follow recent publications in the area (van der Zander et al., 2023; Dörfler et al., 2025) and primarily employ the square-bracket notation.

[2]Interventions are distinct from conditional probability statements such as $\Pr(X = \text{contracted} \mid Y = \text{vaccinated})$. To see this, consider a hypothetical world where the vaccine is ineffective, the disease only exists in a laboratory and an oracle randomly determines whether a person will be infected without vaccination, or receive the vaccine and not come in contact with the disease. In this world, $\Pr(X = \text{contracted} \mid Y = \text{vaccinated}) = 0$ but $\Pr([Y = \text{vaccinated}] \, X = \text{contracted}) > 0$.

dimension to the PCH—specifically, the *breadth* of operations that can be applied to the probability terms themselves. For $\circledast \in \{\mathsf{prob}, \mathsf{causal}, \mathsf{counterfact}\}$, we distinguish the following fragments of the PCH:

- $\mathcal{L}_{\circledast}^{\mathsf{base}}$: only simple probability terms are allowed, such as $\Pr(\cdot) \leq \Pr(\circ)$ or $\Pr(\cdot) \geq 1$;

- $\mathcal{L}_{\circledast}^{\mathsf{lin}}$: linear combinations of probability terms are allowed, such as $\Pr(\cdot) - \Pr(\circ) \geq 3\Pr(\bullet)$;

- $\mathcal{L}_{\circledast}^{\mathsf{poly}}$: polynomials over probability terms are allowed, such as $\Pr(\cdot)^2 \leq 2\Pr(\circ) \cdot \Pr(\bullet) + 0.1$.

Crucially, the various combinations of depth and breadth give rise to a $3 \times 3$ *expressivity matrix* $M$ for PCH (Table 1 in each (van der Zander et al., 2023; Dörfler et al., 2025)).

A crucial and well-studied problem in the setting of causal reasoning is SATISFIABILITY—that is, determining whether a given formula (consisting of a set of probability constraints) admits an SCM (Fagin et al., 1990; Ibeling & Icard, 2020; van der Zander et al., 2023; Mossé et al., 2024; Dörfler et al., 2025). We note that there is a high-level parallel between this SATISFIABILITY problem in the causal setting and the well-known BOOLEAN SATISFIABILITY (SAT) and CONSTRAINT SATISFACTION (CSP) problems; the distinction lies in the types of constraints on the input and the nature of the sought-after model. However, solving SATISFIABILITY in our causal reasoning setting is, in general, a much more daunting task. If we let $\mathsf{SAT}_{\circledast}^*$ denote the SATISFIABILITY problem for formulas from the fragment $\mathcal{L}_{\circledast}^*$ of the PCH, then depending on the choice of $\circledast \in \{\mathsf{prob}, \mathsf{causal}, \mathsf{counterfact}\}$ and $* \in \{\mathsf{base}, \mathsf{lin}, \mathsf{poly}\}$ the problem under consideration will be complete for the complexity classes NP or $\exists\mathbb{R}$—see also the detailed discussion of related work at the end of this section.

Crucially, while previous works have made significant strides towards mapping the classical complexity landscape of the SATISFIABILITY problem, even the "easiest" fragments of the expressivity matrix remain NP-hard. The central aim of this article is to provide a counterweight to this pessimistic perspective and identify fundamental gateways to tractability for SATISFIABILITY, specifically by employing the more refined *parameterized complexity* paradigm (Downey & Fellows, 2013; Cygan et al., 2015). There, one analyzes the running time of algorithms not only in terms of the input size $|I|$, but also with respect to a specified numerical parameter $k$. The standard notion of tractability used in this setting is then tied to algorithms which run in time $f(k) \cdot |I|^{\mathcal{O}(1)}$ for some computable function $f$; problems admitting such *fixed-parameter* algorithms are called *fixed-parameter tractable* (FPT). A weaker—but still useful—notion of tractability stems from the existence of a so-called XP-*algorithm*, i.e., an algorithm running in time $|I|^{f(k)}$ (this gives rise to the complexity class XP). Our

main results include not only the first fixed-parameter and XP-algorithms for the problem, but also matching lower bounds which allow us to identify the limits of parameterized tractability in the expressivity matrix.

**Contributions.** A loose inspiration for this work stems from the success stories in the aforementioned domains of BOOLEAN SATISFIABILITY and CONSTRAINT SATISFACTION. The parameterized complexity of these two problems is by now very well understood, and perhaps the most classical parameterized algorithms use the *primal treewidth* as the parameter of choice. Essentially, this measures how "tree-like" the interactions between the variables in the instance are—more precisely, this is captured by measuring the *treewidth*, a fundamental graph parameter (Robertson & Seymour, 1984), of the graph obtained by representing variables as vertices and using edges to capture the property of lying in the same "term" (i.e., clause or constraint). In particular, it is known that BOOLEAN SATISFIABILITY is fixed-parameter tractable w.r.t. the primal treewidth (Biere et al., 2009, Chapter 13), while CONSTRAINT SATISFACTION admits an XP-algorithm under the same parameterization (Samer & Szeider, 2010); the latter becomes fixed-parameter tractable when the parameterization also includes the domain size for the variables (Samer & Szeider, 2010).

Given the above, it is natural to ask whether one can use the primal treewidth to establish tractability for SATISFIABILITY in the PCH setting. As our first set of contributions, we provide a complete answer to this question: $\mathsf{SAT}_{\mathsf{prob}}^{\mathsf{lin}}$ is

**(1)** in XP w.r.t. the primal treewidth alone, and
**(2)** fixed-parameter tractable w.r.t. the primal treewidth plus the domain size $d$.

Moreover, under well-established complexity assumptions one can neither

**(3)** improve the XP-tractability to FPT (not even for $\mathsf{SAT}_{\mathsf{prob}}^{\mathsf{base}}$), nor
**(4)** lift **any** of these tractability results to $\mathsf{SAT}_{\mathsf{prob}}^{\mathsf{poly}}$ or $\mathsf{SAT}_{\mathsf{causal}}^{\mathsf{lin}}$.

Furthermore, we remark that parameterizing by the domain size alone does not yield tractability under well-established complexity assumptions (see Theorem 4.2).

While the above results are comprehensive, they only provide a gateway to tractability for the "shallowest" probabilistic fragment of the PCH. We hence ask whether one can achieve tractability for deeper fragments of the PCH (that is, $\mathcal{L}_{\mathsf{causal}}^*$ and $\mathcal{L}_{\mathsf{counterfact}}^*$) if the primal treewidth is replaced with a more restrictive parameterization—specifically the number $n$ of variables in the formula. We note that the analogous question in the BOOLEAN SATISFIABILITY and CONSTRAINT SATISFACTION setting is trivial: there, asymptoti-

cally optimal (under well-established complexity assumptions) algorithms parameterized by $n$ can merely enumerate all possible models (Impagliazzo et al., 2001; Karthik C. S. et al., 2024). Such an approach is doomed to fail for the causal SATISFIABILITY problem: not only will an SCM contain (potentially many) auxiliary random variables, but also variable dependencies and random distributions that cannot be exhaustively enumerated.

As our second set of contributions, we map the complexity landscape for deeper fragments of the PCH as well: $\text{SAT}^{\text{lin}}_{\text{counterfact}}$ is

(5) fixed-parameter tractable w.r.t. $n$ plus domain size $d$.

Moreover, under well-established complexity assumptions one can neither

(6) drop any of the parameters while preserving fixed-parameter tractability (not even for $\text{SAT}^{\text{base}}_{\text{prob}}$), nor

(7) lift the fixed-parameter tractability to $\text{SAT}^{\text{poly}}_{\text{counterfact}}$.

A schematic overview of our contributions is provided in the mind map below (Figure 1).

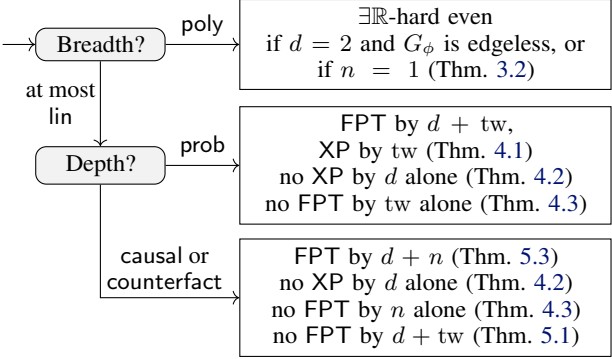

*Figure 1.* Parameterized complexity of SATISFIABILITY in the PCH based on the position (breadth/depth) in the expressivity matrix $M$. All results hold under well-established complexity assumptions and refer to an instance with $n$ observed variables over a domain of size $d$ such that the treewidth of the primal graph $G_\phi$ is tw. We note that the $\exists\mathbb{R}$-hardness of the poly fragment was already established by Mossé et al. (2024), but not under the stated restrictions which rule out tractability in the parameterized setting.

**Extensions to Marginalization.** In order to efficiently express marginalization, recent works (van der Zander et al., 2023; Dörfler et al., 2025) have extended the classical fragments in $M$ to $\mathcal{L}^{\text{base}\langle\Sigma\rangle}_{\circledast}$, $\mathcal{L}^{\text{lin}\langle\Sigma\rangle}_{\circledast}$, and $\mathcal{L}^{\text{poly}\langle\Sigma\rangle}_{\circledast}$, respectively; the only difference is that these classes additionally include the unary summation operator $\sum$. Depending on the specific fragment considered, including these marginalization operators in the SATISFIABILITY problem yields completeness for the complexity classes $\text{NP}^{\text{PP}}$, PSPACE, NEXP or succ-$\exists\mathbb{R}$—see Table 1 in (Dörfler et al., 2025). Since the algorithm underlying Result (5) can also be used to es-

tablish inclusion in the complexity class EXPTIME while $\text{SAT}^{\text{lin}\langle\Sigma\rangle}_{\text{counterfact}}$ is NEXP-complete (Dörfler et al., 2025), under well-established complexity assumptions it is not possible to lift our results towards full marginalization operators as considered in the aforementioned works. Nevertheless, if one were to bound the nesting depth of the unary summation operator $\sum$ to any (arbitrary but fixed) constant, all of our results could be directly translated to the marginalization setting by simply expanding on the respective sums.

**Proof Techniques.** The standard approach to establishing tractability for problems parameterized by treewidth is to employ dynamic programming—this is the approach used not only for the aforementioned treewidth-based algorithms solving BOOLEAN SATISFIABILITY and CONSTRAINT SATISFACTION, but also for almost every algorithm parameterized by treewidth. From a technical standpoint, it is hence surprising that our results **do not** employ dynamic programming at all; in fact, the SATISFIABILITY problem seems entirely incompatible with the usual "leaf-to-root" dynamic programming paradigm used for treewidth.

Instead, our proof of Results (1) and (2) relies on an entirely novel approach. We first prove that every YES-instance of $\text{SAT}^{\text{lin}}_{\text{prob}}$ with primal treewidth $k$ admits a "well-structured" SCM whose hidden variables and dependencies can be neatly mapped onto the tree-like structure of the primal graph and determined in advance. However, this step on its own cannot determine whether an SCM actually exists, as for that we need to compute and verify the probability distributions for the hidden variables. In the second step, we use the tree-likeness of the instance once again to construct a "fixed-parameter sized" linear program which either computes a viable set of probability distributions, or determines that none exists. It is well-known that linear programs can be solved in polynomial time (Papadimitriou & Steiglitz, 1998)—the difficult part lies in building a program that provably verifies the existence of an SCM while avoiding an exponential dependency on the input size.

In order to apply the reduction technique to Result (5), we need to be able to deal with the presence of interventions in the formula. Towards this, we argue that every YES-instance of $\text{SAT}^{\text{lin}}_{\text{counterfact}}$ admits an SCM with different structural properties than those used for Result (5): in particular, the value of a single hidden variable $U$ determines not just the value of each observed variable but the *function* of how it is determined from the other observed variables. We then define a suitable linear programming formulation that targets the computation of such well-structured SCMs.

For establishing the lower-bound results, we develop three distinct reductions: one from MULTICOLORED-CLIQUE which handles (3) and (6), one from a restricted variant of the Existential Theory of the Reals problem for Results

**(4)** and **(7)**, and a separate reduction from 3-SAT for the remaining lin-causal case of **(4)**.

**Related Work.** $\mathrm{SAT}_{\mathsf{prob}}^{\mathsf{lin}}$ and $\mathrm{SAT}_{\mathsf{prob}}^{\mathsf{base}}$ were shown to be NP-complete by Fagin et al. (1990), and analogous results for the fragments $\mathrm{SAT}_{\mathsf{causal}}^{\mathsf{lin}}$, $\mathrm{SAT}_{\mathsf{causal}}^{\mathsf{base}}$, $\mathrm{SAT}_{\mathsf{counterfact}}^{\mathsf{lin}}$, and $\mathrm{SAT}_{\mathsf{counterfact}}^{\mathsf{base}}$ were obtained by Mossé et al. (2024). The $\exists\mathbb{R}$-completeness of $\mathrm{SAT}_{\mathsf{prob}}^{\mathsf{poly}}$, $\mathrm{SAT}_{\mathsf{causal}}^{\mathsf{poly}}$ and $\mathrm{SAT}_{\mathsf{counterfact}}^{\mathsf{poly}}$ was also established in the latter work (Mossé et al., 2024). As mentioned above, these complexity-theoretic studies were recently extended to languages containing the summation operator $\sum$ (van der Zander et al., 2023; Dörfler et al., 2025). Other related languages designed to express probabilistic reasoning were developed in the works of, e.g., Nilsson (1986); Georgakopoulos et al. (1988); Ibeling & Icard (2020). Moreover, the existence of solutions to the SATISFIABILITY problem with specific properties has very recently been studied by Bläser et al. (2025).

As demonstrated by van der Zander et al. (2023), causal identification as well as enforcing other properties of SCMs can be phrased as a question of satisfiability in the PCH. Conceptually, one can express graph properties via additional constraints with interventions.

Beyond SATISFIABILITY, the parameterized complexity paradigm has been employed in several works studying another central problem in the area of causality: BAYESIAN NETWORK STRUCTURE LEARNING. This line of research was initiated by Ordyniak & Szeider (2013), with recent contributions considering a broad range of parameterizations and variations of the problem (Ganian & Korchemna, 2021; Grüttemeier et al., 2021a;b; Grüttemeier & Komusiewicz, 2022). The complexity of the related CAUSAL DISCOVERY problem was recently studied by Ganian et al. (2024).

Beyond the aforementioned prominent applications in BOOLEAN SATISFIABILITY and CONSTRAINT SATISFACTION, primal treewidth has been used as a natural means of capturing structural properties of inputs in a variety of other settings. Examples of this in the broad AI area include its applications in INTEGER LINEAR PROGRAMMING (Ganian et al., 2017; Ganian & Ordyniak, 2018), HEDONIC GAMES (Peters, 2016; Hanaka & Lampis, 2022), MATRIX COMPLETION (Ganian et al., 2022), ANSWER SET PROGRAMMING (Fichte & Hecher, 2018), and RESOURCE ALLOCATION (Eiben et al., 2023). We note that the treewidth-based algorithms in all of the aforementioned works rely on dynamic programming, which fundamentally differs from the technique employed to achieve our Results **(1)** and **(2)**.

## 2. Preliminaries

For $n \in \mathbb{N}$, let $[n] = \{1, \ldots, n\}$. For $i_1, i_2 \in \mathbb{R}$, let $[i_1, i_2] = \{j \in \mathbb{R} \mid i_1 \leq j \leq i_2\}$. We follow established

notation as used in (Mossé et al., 2024; van der Zander et al., 2023). Let $\mathbf{V}$ be a set of random variables and, w.l.o.g., assume that they share a given domain $D$ of size $d$. We note that throughout this paper, we assume random variables to range over discrete options (i.e., we effectively consider discrete probability distributions). While we do not formally define the complexity class W[1], it is well-known that hardness for the class effectively excludes fixed-parameter tractability (Downey & Fellows, 2013; Cygan et al., 2015).

**Syntax of the Languages of PCH.** For $V \in \mathbf{V}$ and $v \in D$, a statement $V = v$ is called an *atom*. *Events* over $\mathbf{V}$ are combinations of atoms following certain grammatical rules (where $\top$ denotes the empty intervention):

$$\mathcal{E}_{\mathsf{prop}} ::= V = v \mid \neg \mathcal{E}_{\mathsf{prop}} \mid \mathcal{E}_{\mathsf{prop}} \wedge \mathcal{E}_{\mathsf{prop}},$$
$$\mathcal{E}_{\mathsf{int}} ::= \top \mid V = v \mid \mathcal{E}_{\mathsf{int}} \wedge \mathcal{E}_{\mathsf{int}},$$
$$\mathcal{E}_{\mathsf{post\text{-}int}} ::= [\mathcal{E}_{\mathsf{int}}] \, \mathcal{E}_{\mathsf{prop}},$$
$$\mathcal{E}_{\mathsf{counterf}} ::= \mathcal{E}_{\mathsf{post\text{-}int}} \mid \neg \mathcal{E}_{\mathsf{counterf}} \mid \mathcal{E}_{\mathsf{counterf}} \wedge \mathcal{E}_{\mathsf{counterf}}.$$

We call events in $\mathcal{E}_{\mathsf{prop}}$ *propositions* and events in $\mathcal{E}_{\mathsf{int}}$ *interventions*. Each event $\varepsilon$ can only occur within a probabilistic statement $\mathrm{Pr}(\varepsilon)$, called a *term*. The *size of a term* is the number of its atoms. For $\mathcal{E} \in \{\mathcal{E}_{\mathsf{prop}}, \mathcal{E}_{\mathsf{post\text{-}int}}, \mathcal{E}_{\mathsf{counterf}}\}$ and $\varepsilon \in \mathcal{E}$, we define the following ways of combining terms.

$$T_{\mathsf{base}}(\mathcal{E}) ::= \mathrm{Pr}(\varepsilon),$$
$$T_{\mathsf{lin}}(\mathcal{E}) ::= \mathrm{Pr}(\varepsilon) \mid T_{\mathsf{lin}}(\mathcal{E}) + T_{\mathsf{lin}}(\mathcal{E}),$$
$$T_{\mathsf{poly}}(\mathcal{E}) ::= \mathrm{Pr}(\varepsilon) \mid T_{\mathsf{poly}}(\mathcal{E}) + T_{\mathsf{poly}}(\mathcal{E}) \mid$$
$$T_{\mathsf{poly}}(\mathcal{E}) \cdot T_{\mathsf{poly}}(\mathcal{E}).$$

Last, for $* \in \{\mathsf{base}, \mathsf{lin}, \mathsf{poly}\}$ we define $\mathcal{L}_{\mathsf{prob}}^*$, $\mathcal{L}_{\mathsf{causal}}^*$, and $\mathcal{L}_{\mathsf{counterfact}}^*$ to be the *languages* that contain all sets of inequalities over elements in $T_*(\mathcal{E}_{\mathsf{prop}})$, $T_*(\mathcal{E}_{\mathsf{post\text{-}int}})$, and $T_*(\mathcal{E}_{\mathsf{counterf}})$, respectively. The elements inside $\mathcal{L}_{\mathsf{prob}}^*$, $\mathcal{L}_{\mathsf{causal}}^*$, and $\mathcal{L}_{\mathsf{counterfact}}^*$ are called *formulas*. Note that tautological and contradictory events can be used to encode comparisons against 1 and 0, such as $\mathrm{Pr}(\varepsilon) \leq 0$. Moreover, the grammars of $\mathcal{L}_{\circledast}^{\mathsf{lin}}$ and $\mathcal{L}_{\circledast}^{\mathsf{poly}}$ support integer coefficients, which can be effectively constructed by summing up multiple probabilities of the same type. Any inequality with rational coefficients can be encoded by multiplying both sides with the smallest common multiple of all non-integer coefficients. At the beginning of the next section, we will compare our syntax to the one used in related work.

**Semantics of the Languages of PCH.** We define the semantics of the aforementioned languages using the notion of Structural Causal Models as popularized by Glymour et al. (1987) and Pearl (2009, Section 3.2). A *recursive Structural Causal Model* (*SCM*, or simply *model*) over domain $D$ is a tuple $\mathcal{M} = (\mathbf{V}, \mathbf{U}, \mathcal{F}, \mathbb{P})$ with

- a set $\mathbf{V}$ of endogenous (observed) variables, implicitly well-ordered by $\prec$, that range over $D$,

- a set $\mathbf{U}$ of exogenous (hidden) variables,

- a set $\mathcal{F} = \{f_V\}_{V \in \mathbf{V}}$ of functions where $f_V$ specifies how the value of $V$ is determined from $\mathbf{U}$ and $\mathbf{V}_{\prec V}$, that is, the subset of $\mathbf{V}$ that precedes $V \in \mathbf{V}$ in $\prec$,

- a probability distribution $\mathbb{P}$ on $\mathbf{U}$.

Any model $\mathcal{M}$ whose exogenous variables $\mathbf{U}$ have an infinite or continuous domain is (w.r.t. its evaluation) equivalent to a model $\mathcal{M}'$ where all exogenous variables have discrete and finite domains (Zhang et al., 2022). Consequently, we assume throughout that each variable $U \in \mathbf{U}$ has a discrete and finite domain $\mathrm{Val}(U)$, and let $\mathrm{Val}(\mathbf{U}) = \mathrm{Val}(U_1) \times \ldots \times \mathrm{Val}(U_{|\mathbf{U}|})$ refer to their combined range.

Let $V = v$ be an atom in $\mathcal{E}_{\mathsf{int}}$. We denote by $\mathcal{F}_{V=v}$ the set of functions obtained from $\mathcal{F}$ by replacing $f_V$ with the constant function $v$. We generalize this definition to arbitrary conjunctions of atoms $\gamma \in \mathcal{E}_{\mathsf{int}}$ in the natural way and denote the set of resulting functions as $\mathcal{F}_\gamma$. Let $\varepsilon \in \mathcal{E}_{\mathsf{prop}}$ and $\overline{u} \in \mathrm{Val}(\mathbf{U})$. We write $\mathcal{F}, \overline{u} \models \varepsilon$ if evaluating $\mathcal{F}$ on input $\overline{u}$ yields an assignment to $\mathbf{V}$ under which $\varepsilon$ is satisfied. For $[\gamma]\,\varepsilon \in \mathcal{E}_{\mathsf{post\text{-}int}}$, we write $\mathcal{F}, \overline{u} \models [\gamma]\,\varepsilon$ if $\mathcal{F}_\gamma, \overline{u} \models \varepsilon$. Moreover, for all $\varepsilon, \varepsilon_1, \varepsilon_2 \in \mathcal{E}_{\mathsf{counterf}}$, we write (i) $\mathcal{F}, \overline{u} \models \neg\varepsilon$ if $\mathcal{F}, \overline{u} \not\models \varepsilon$, and (ii) $\mathcal{F}, \overline{u} \models \varepsilon_1 \wedge \varepsilon_2$ if both $\mathcal{F}, \overline{u} \models \varepsilon_1$ and $\mathcal{F}, \overline{u} \models \varepsilon_2$. For a given model $\mathcal{M} = (\mathbf{V}, \mathbf{U}, \mathcal{F}, \mathbb{P})$, we denote $S_{\mathcal{M}}(\varepsilon) := \{\overline{u} \in \mathrm{Val}(\mathbf{U}) \mid \mathcal{F}, \overline{u} \models \varepsilon\}$. The way $\mathcal{M}$ interprets an expression $\mathbf{t} \in T_{\mathsf{poly}}(\mathcal{E})$ is denoted by $[\![\mathbf{t}]\!]_{\mathcal{M}}$ and recursively defined as follows: $[\![\mathrm{Pr}(\varepsilon)]\!]_{\mathcal{M}} = \sum_{\overline{u} \in S_{\mathcal{M}}(\varepsilon)} \mathbb{P}(\overline{u})$. For two expressions $\mathbf{t}_1, \mathbf{t}_2 \in T_{\mathsf{poly}}(\mathcal{E})$, we define $\mathcal{M} \models \mathbf{t}_1 \leq \mathbf{t}_2$ if and only if $[\![\mathbf{t}_1]\!]_{\mathcal{M}} \leq [\![\mathbf{t}_2]\!]_{\mathcal{M}}$. Negation and conjunction are defined in the usual way, yielding the semantics for $\mathcal{M} \models \phi$ for any formula $\phi \in \mathcal{L}_{\mathsf{counterfact}}^{\mathsf{poly}}$.

**Primal Treewidth.** Let $\phi \in \mathcal{L}_{\mathsf{counterfact}}^{\mathsf{poly}}$ be a formula over variables $\mathbf{V}$. By $G_\phi = (\mathbf{V}, E)$, we denote the *primal graph* of $\phi$, where $\{V, V'\} \in E$ if and only if $V \neq V'$ and there is a term in $\phi$ containing both $V$ and $V'$. The *treewidth* $\mathrm{tw}(G)$ of a graph $G$ is a well-established measure of how "tree-like" it is; for instance, trees have a treewidth of $1$, while an $s$-vertex complete graph has treewidth $s - 1$.

Formally, a *nice tree decomposition* of $G_\phi$ is a pair $(\mathbf{T}, \chi)$, where $\mathbf{T}$ is a tree (whose vertices are called *nodes*) rooted at a node $N_0$ and $\chi$ is a function that assigns to each node $N$ a set $\chi(N) \subseteq \mathbf{V}$ such that:

- For every $\{V, V'\} \in E$, there is a node $N$ such that $\{V, V'\} \subseteq \chi(N)$.

- For every vertex $V \in \mathbf{V}$, the set of nodes $N$ satisfying $V \in \chi(N)$ forms a subtree of $\mathbf{T}$.

- $|\chi(N)| = 0$ if $N$ is a leaf of $\mathbf{T}$ or $N = N_0$.

- There are only three kinds of non-leaf nodes in $\mathbf{T}$:

  - *introduce*: a node $N$ with exactly one child $N'$ such that $\chi(N) = \chi(N') \cup \{V\}$ for a $V \notin \chi(N')$.
  - *forget*: a node $N$ with exactly one child $N'$ such that $\chi(N) = \chi(N') \setminus \{V\}$ for a $V \in \chi(N')$.
  - *join*: a node $N$ with two children $N_1, N_2$ such that $\chi(N) = \chi(N_1) = \chi(N_2)$.

We call each set $\chi(N)$ a *bag*. The width of a nice tree decomposition $(\mathbf{T}, \chi)$ is the size of the largest bag $\chi(N)$ minus 1, and the *treewidth of $G_\phi$*, denoted by $\mathrm{tw}(G_\phi)$, is the minimum width of a nice tree decomposition of $G_\phi$.

We let the (*primal*) *treewidth of a formula* $\phi$ denote the treewidth of its primal graph, that is, $\mathrm{tw}(\phi) = \mathrm{tw}(G_\phi)$.

# 3. Satisfiability for Languages of PCH and Structural Insights

We examine several analogues of the well-known problem BOOLEAN SATISFIABILITY that capture various probabilistic, causal, and counterfactual statements. We denote these problems as $\mathrm{SAT}_{\circledast}^*$, where $\circledast \in \{\mathsf{prob}, \mathsf{causal}, \mathsf{counterfact}\}$ and $* \in \{\mathsf{base}, \mathsf{lin}, \mathsf{poly}\}$, and define them as follows.

| $\mathrm{SAT}_{\circledast}^*$ | |
| --- | --- |
| **Input:** | A set $D$ of $d$ domain values and a formula $\phi \in \mathcal{L}_{\circledast}^*$ over variables $\mathbf{V} = \{V_1, \ldots, V_n\}$. |
| **Task:** | Decide if there exists a recursive Structural Causal Model $\mathcal{M}$ over $D$ such that $\mathcal{M} \models \phi$. |

The classical computational complexity of $\mathrm{SAT}_{\circledast}^*$ has by now been studied extensively (Fagin et al., 1990; Ibeling & Icard, 2020; van der Zander et al., 2023; Mossé et al., 2024; Dörfler et al., 2025). We remark that our definition of $\mathrm{SAT}_{\circledast}^*$ slightly deviates from the one established in previous works, in the sense that we restrict our attention to input formulas $\phi$ that are *sets of inequalities* (that is, each inequality forms a constraint) rather than allowing arbitrary Boolean combinations of inequalities. However, this restriction does not affect any of the known complexity-theoretic results, since previous lower-bound proofs did not employ any Boolean combinations beyond sets. However, the situation changes drastically when studying $\mathrm{SAT}_{\circledast}^*$ from the viewpoint of parameterized complexity, as we show next.

Let $\mathrm{arbSAT}_{\mathsf{prob}}^{\mathsf{base}}$ denote the version of $\mathrm{SAT}_{\mathsf{prob}}^{\mathsf{base}}$ in which $\phi$ is an arbitrary Boolean combination of inequalities over elements in $T_{\mathsf{base}}(\mathcal{E}_{\mathsf{prop}})$. We justify our restriction to $\mathrm{SAT}_{\circledast}^*$ by the hardness of $\mathrm{arbSAT}_{\mathsf{prob}}^{\mathsf{base}}$ in a very restricted setting, thus dashing any hope to exploit structural properties of $\phi$.

**Theorem 3.1.** $\mathrm{arbSAT}_{\mathsf{prob}}^{\mathsf{base}}$ *is* NP*-complete even if $G_\phi$ is edgeless and $d = 2$.*

*Proof Sketch.* We extend the NP-completeness of $\mathrm{arbSAT}^{\mathsf{base}}_{\mathsf{prob}}$ (Fagin et al., 1990) to the stated restricted case via a reduction from 3-SAT. Each 3-SAT-variable $x$ becomes a variable $X$ in $\mathbf{V}$, constraints ensure that each variable is always 1 or always 0 and a constraint captures all clauses in the full 3-SAT formula, where literals $x$ and $\bar{x}$ become $\Pr(X = 1)$ and $\Pr(X = 0)$, respectively. $\qquad\square$

**Intractability of $\mathrm{SAT}^{\mathsf{poly}}_{\mathsf{prob}}$.** Our main contributions target the lin and base fragments of the expressivity matrix. Here, we show that the tractability results obtained in Sections 4 and 5 most likely not hold for polynomial inequalities. The proof is based on a reduction from the $\exists\mathbb{R}$-complete problem $\mathrm{ETR}^{1/8,+,\times}_{[-1/8,1/8]}$ (Abrahamsen et al., 2022), where the probabilities of individual events are used to encode the values of the variables in a way that conceptually resembles a construction by van der Zander et al. (2023, Prop. 6.5).

**Theorem 3.2.** $\mathrm{SAT}^{\mathsf{poly}}_{\mathsf{prob}}$ *is $\exists\mathbb{R}$-complete even if $n = 1$, or if $d = 2$ and $G_\phi$ is edgeless.*

**Further Structural Insights in $\mathrm{SAT}^*_\circledast$.** In order to facilitate our complexity-theoretic analysis, we emphasize that a Structural Causal Model can be efficiently evaluated, that is, given the values of its hidden variables, it can be decided in polynomial time, whether a certain event happens.

**Observation 3.3.** *Given a model $\mathcal{M} = (\mathbf{U}, \mathbf{V}, \mathcal{F}, \mathbb{P})$, an event $\varepsilon \in \mathcal{E}_{\mathsf{counterf}}$, and some $\bar{u} \in \mathrm{Val}(\mathbf{U})$, let $|\varepsilon|$ denote the number of atoms in $\varepsilon$. Assuming that each function in $\mathcal{F}$ can be evaluated in time $\mathcal{O}(n)$, one can decide whether $\mathcal{F}, \bar{u} \models \varepsilon$ in time in $\mathcal{O}(n^2 + |\varepsilon|)$.*

The running time of our algorithmic results often depends on the size $d$ of the domain $D$. Assuming that $d$ is not much larger than the size of $\phi$ does not reduce the generality of our results, as we can always reduce to an equivalent instance where $d$ is bounded from above by $|\phi| + 1$.

**Observation 3.4.** *Consider an instance of $\mathrm{SAT}^*_\circledast$ consisting of a domain $D$ and a formula $\phi \in \mathcal{L}^*_\circledast$. Let $D_\phi$ be the set of values in $D$ that are explicitly mentioned in at least one atom in $\phi$ and choose some $\gamma \notin D_\phi$. Then, it holds that $\phi$ over domain $D_\phi \cup \{\gamma\}$ is a YES-instance of $\mathrm{SAT}^*_\circledast$ if and only if so is $\phi$ over $D$.*

## 4. Linear Inequalities over Probabilistic Expressions

This section is dedicated to the complexity-theoretic analysis of $\mathrm{SAT}^*_{\mathsf{prob}}$, that is, the satisfiability problem for the layer of the PCH that does not allow any interventions. First, we establish the main tractability result of this section, and then proceed by showing that it is tight as outlined in Figure 1.

**Theorem 4.1.** $\mathrm{SAT}^{\mathsf{lin}}_{\mathsf{prob}}$ *is in* FPT *w.r.t. the combined parameter $d + \mathrm{tw}(\phi)$, and in* XP *w.r.t. $\mathrm{tw}(\phi)$.*

*Proof.* Consider an instance of $\mathrm{SAT}^{\mathsf{lin}}_{\mathsf{prob}}$ with formula $\phi$ and domain $D$. We prove both statements simultaneously by describing an algorithm that runs in time $d^{f(\mathrm{tw}(\phi))}|\phi|^{\mathcal{O}(1)}$, for a computable function $f$. Consider a nice tree decomposition of $G_\phi$ consisting of $\mathcal{O}(n)$ nodes with maximum size $w := \mathrm{tw}(\phi) + 1$ computed by, e.g., the algorithm of (Bodlaender, 1996). Without loss of generality, assume that only the bags of leaf nodes are empty and ignore them in the following procedure (instead we consider their parents as leaves). For the remaining tree decomposition $\mathbf{T}$, let $D^{|B|}$ be the combined domain of the variables of bag $B$ in $\mathbf{T}$. We construct the following Linear Program (LP). For each distinct bag $B$ and $\bar{v} \in D^{|B|}$, construct an LP-variable $p_{B=\bar{v}}$ (note that distinct nodes might share a bag). This will capture the probability of the event $B = \bar{v}$, that is, each variable in $B$ takes the respective value in $\bar{v}$. To ensure a valid probability distribution over the LP-variables in each bag $B$, add the LP-constraints

$$p_{B=\bar{v}} \geq 0 \quad \text{for each LP-variable } p_{B=\bar{v}}, \quad \text{and}$$
$$\sum\nolimits_{\bar{v} \in D^{|B|}} p_{B=\bar{v}} = 1 \quad \text{for each bag } B.$$

For every pair of distinct bags $B, B'$ whose nodes are adjacent in $\mathbf{T}$ and $B \neq B'$, note that there is some $V \in \mathbf{V}$ such that, without loss of generality, $B' = B \cup \{V\}$. To guarantee consistency between the probability distributions of $B$ and $B'$, we add for each such pair and each $\bar{v} \in D^{|B|}$ the LP-constraint

$$p_{B=\bar{v}} = \mathrm{sum}\{p_{B'=\bar{v}'} \mid \bar{v}' \in D^{|B'|} \text{ and } \bar{v}' \text{ sets } B \text{ to } \bar{v}\}.$$

Next, for each constraint $C$ in $\phi$, consider each of its terms $\Pr(\varepsilon)$ and define $\mathcal{V}_\varepsilon \subseteq \mathbf{V}$ to be the set of variables that occur in $\Pr(\varepsilon)$. By construction, for each $\varepsilon$, all variables in $\mathcal{V}_\varepsilon$ form a clique in $G_\phi$. Consequently, there is at least one bag $B_\varepsilon$ in $\mathbf{T}$ such that $\mathcal{V}_\varepsilon \subseteq B_\varepsilon$. Consider an arbitrary choice of such $B_\varepsilon$ and obtain an LP-constraint from $C$ by replacing each occurrence of term $\Pr(\varepsilon)$ by a sum over all LP-variables $p_{B_\varepsilon=\bar{v}}$ such that $B_\varepsilon = \bar{v}$ satisfies the event $\varepsilon$. Then the LP consists of $\mathcal{O}(n \cdot d^w)$ LP-variables and $\mathcal{O}(|\phi| + n \cdot d^w)$ LP-constraints.

We can find a solution of an LP (or decide that there is none) in polynomial time with respect to its size, that is, the number of its variables plus constraints. Crucially, if $\phi$ is a YES-instance witnessed by a model $\mathcal{M}$ which induces a probability distribution over $\mathbf{V}$, then the LP admits a solution; indeed, we can satisfy all constraints by setting each LP-variable $p_{B=\bar{v}}$ to the probability of the event $B = \bar{v}$ within that distribution.

For the converse, assume the LP has a solution. We construct a model for $\phi$ by passing through $\mathbf{T}$ in a breadth-first-search manner, starting from an arbitrary leaf node with some bag $B = \{V\}$. Let $U_V$ be a hidden variable with domain $D$

such that $\mathbb{P}(U_V = v) = p_{B=(v)}$ for all $v \in D$. Whenever we transition from a bag $B$ to a bag $B'$ containing a variable $V$ which we have not yet described in our model, we have $B' = B \cup \{V\}$. For each $\overline{v} \in D^{|B|}$ such that $p_{B=\overline{v}} > 0$, create a hidden variable $U_{V|B=\overline{v}}$ with domain $\mathrm{Val}(U_{V|B=\overline{v}}) = D$ and let

$$\mathbb{P}(U_{V|B=\overline{v}} = x) := \frac{p_{B'=\overline{v}+x}}{p_{B=\overline{v}}} \quad \text{for each } x \in D,$$

where $(\overline{v} + x) \in D^{|B'|}$ is such that it sets $B$ to $\overline{v}$ and $V$ to $x$. This describes a valid probability distribution of $U_{V|B=\overline{v}}$: As $\mathbb{P}(U_{V|B=\overline{v}} = x) \geq 0$ for all $x$, it remains to show that $\sum_{x \in D} \mathbb{P}(U_{V|B=\overline{v}} = x) = 1$. We have

$$\sum_{x \in D} \mathbb{P}(U_{V|B=\overline{v}} = x) = \sum_{x \in D} \frac{p_{B'=\overline{v}+x}}{p_{B=\overline{v}}}$$
$$= \frac{1}{p_{B=\overline{v}}} \sum_{x \in D} p_{B'=\overline{v}+x} = 1,$$

as we ensured $p_{B=\overline{v}} = \sum_{x \in D} p_{B'=\overline{v}+x}$ by an LP-constraint. We now define the function $f_V$ such that, for each $\overline{v} \in D^{|B|}$, if $B = \overline{v}$ then $V = U_{V|B=\overline{v}}$.

It remains to argue that the model $\mathcal{M}$ obtained after visiting every node witnesses $\phi$ to be a YES-instance. To this end, employ induction over the breadth-first search described above to prove that after visiting a node with bag $B$, for each $\overline{v} \in D^{|B|}$ the value of $p_{B=\overline{v}}$ describes the probability of $B = \overline{v}$ in the current model $\mathcal{M}'$, that is, $[\![\Pr(B = \overline{v})]\!]_{\mathcal{M}'} = p_{B=\overline{v}}$. As the bag of the first node contains just one variable, the base case is trivial. Now assume that the claim holds for bag $B$ of some node $N$ and model $\mathcal{M}$, and from $N$ we are visiting a node $N'$ with bag $B'$. If $B' \subseteq B$, then $\mathcal{M}$ is not changed and the claim follows immediately. Otherwise $B' = B \cup \{V\}$ for a variable $V$ and $\mathcal{M}$ is extended to a model $\mathcal{M}'$ as described above. Consider any $\overline{v}' \in D^{|B'|}$. Let $\overline{v}$ equal $\overline{v}'$ when restricted to the variables in $B$ and let $x$ be the value of variable $V$ in $\overline{v}'$. By the construction of $\mathcal{M}'$ and the induction hypothesis, we have that $[\![\Pr(B = \overline{v})]\!]_{\mathcal{M}'} = [\![\Pr(B = \overline{v})]\!]_{\mathcal{M}} = p_{B=\overline{v}}$. If $[\![\Pr(B = \overline{v})]\!]_{\mathcal{M}'} = 0$, then $[\![\Pr(B' = \overline{v}')]\!]_{\mathcal{M}'} = 0$, which is correct by the consistency constraints $p_{B'=\overline{v}'} \leq p_{B=\overline{v}} = 0$. Otherwise,

$$[\![\Pr(B' = \overline{v}')]\!]_{\mathcal{M}'} = [\![\Pr(B = \overline{v})]\!]_{\mathcal{M}'} \cdot \mathbb{P}(U_{V|B=\overline{v}} = x)$$
$$= p_{B=\overline{v}} \cdot \frac{p_{B'=\overline{v}'}}{p_{B=\overline{v}}} = p_{B'=\overline{v}'}.$$

Given a nice tree decomposition, the LP can be constructed and solved in time in $(|\phi| + n \cdot d^w)^{\mathcal{O}(1)}$. For YES-instances, this time also suffices to construct a suitable model.

$\square$

We now show that under well-established complexity assumptions, parameterization by $d$ alone cannot yield tractability, even if the primal graph $G_\phi$ has bounded degree.

**Theorem 4.2.** $\mathrm{SAT}_{\mathrm{prob}}^{\mathrm{base}}$ *is NP-complete even if $d = 2$ and the maximum degree of $G_\phi$ is 8.*

*Proof Sketch.* The containment in NP follows from $\mathrm{arbSAT}_{\mathrm{prob}}^{\mathrm{base}} \in \mathrm{NP}$ (Fagin et al., 1990). To show hardness in our restricted setting, we perform a reduction from 3-SAT. Note that 3-SAT remains NP-hard when restricted to formulas in which each variable occurs exactly twice negated and twice non-negated (Darmann & Döcker, 2021). Thus, w.l.o.g., we assume that our formula $\Phi := \bigwedge_i C_i$ with $C_i := \bigvee_{j \in [3]} \ell_{i_j}$ has this property. We construct an instance $\phi$ of $\mathrm{SAT}_{\mathrm{prob}}^{\mathrm{base}}$ with $\mathbf{V} = \{V_v \mid v \in \mathcal{V}\}$ and $D = \{0, 1\}$ such that for each $C_i \in \Phi$, the constraint $\Pr(\bigvee_{j \in [3]} g(\ell_{i_j})) = 1$ is added to $\phi$, where $g(\ell_{i_j})$ is replaced by $V_v = 1$ if $\ell_{i_j} = v$, and by $V_v = 0$ if $\ell_{i_j} = \neg v$. Since each variable occurs in at most 4 clauses, there are at most 8 other variables it co-occurs with; consequently, $G_\phi$ has a maximum degree of 8. It remains to argue that $\Phi$ is satisfied if and only if $\phi$ admits an SCM. For this, the crucial insight is that the values of the endogenous variables in any SCM that satisfies $\phi$ correspond to a satisfying assignment of the variables in $\Phi$. $\square$

The following result complements Theorem 4.2 by ruling out fixed-parameter tractable algorithms for $\mathrm{SAT}_{\mathrm{prob}}^{\mathrm{base}}$ under a different parameterization, namely the number $n$ of variables. Note that since $\mathrm{tw}(\phi) \leq n$, this implies that we should not expect the primal treewidth of a graph to yield fixed-parameter tractability for $\mathrm{SAT}_{\mathrm{prob}}^{\mathrm{base}}$ alone.

**Theorem 4.3.** $\mathrm{SAT}_{\mathrm{prob}}^{\mathrm{base}}$ *is W[1]-hard w.r.t. $n$.*

*Proof Sketch.* We reduce from MULTICOLORED-CLIQUE, which asks, given a properly vertex-colored graph $G$ with colors $1, \ldots, k$ and vertices $v_1, \ldots, v_r$, whether $G$ contains a $k$-clique. Given $G$, we construct an instance $\phi$ of $\mathrm{SAT}_{\mathrm{prob}}^{\mathrm{base}}$ as follows. Let $\mathbf{V} = \{V_1, \ldots, V_k\}$ and $D = \{v_1, \ldots, v_r\}$. For each $i \in [k]$ and $a \in [r]$, add the constraint $\Pr(V_i = v_a) \leq 0$, unless $v_a$ has color $i$. For each non-adjacent $v_a, v_b$ with $a < b$ and colors $i$, $j$, add the constraint $\Pr(V_i = v_a \wedge V_j = v_b) \leq 0$. The construction takes polynomial time and sets $n = k$. It remains to show that $G$ contains a $k$-clique if and only if $\phi$ admits an SCM. $\square$

## 5. Linear Inequalities over Causal or Counterfactual Expressions

In this section, we turn our attention to interventional causal reasoning. We initiate our study by showing that the FPT-tractability that was established in Theorem 4.1 does not carry over.

**Theorem 5.1.** $\mathrm{SAT}_{\mathrm{causal}}^{\mathrm{lin}}$ *is NP-complete even if $d = 2$ and all edges in $G_\phi$ are pairwise vertex-disjoint.*

*Proof Sketch.* Fagin et al. (1990) showed that arbSAT$^{\text{lin}}_{\text{causal}}$ is NP-complete. To show NP-hardness even under the stated restrictions, we reduce from 3-SAT. Consider a 3-SAT formula $\Phi$ with $r$ variables. We define an instance $\phi$ of SAT$^{\text{lin}}_{\text{causal}}$ with domain $D = \{0, 1\}$ as follows. For each variable $x_i \in \Phi$, we introduce endogenous variables $V_i$ and $\overline{V}_i$, as well as the constraints

$$\Pr([V_i = 1]\,\overline{V}_i = 1) = 0, \text{ and } \Pr([\overline{V}_i = 1]\,V_i = 1) = 0.$$

Furthermore, for each clause $\ell_1 \vee \ell_2 \vee \ell_3$ in $\Phi$, we add $\Pr(L_1 = 1) + \Pr(L_2 = 1) + \Pr(L_3 = 1) \geq 1$ to $\phi$, where, $L_j = V_i$ if $\ell_j = x_i$, and $L_j = \overline{V}_i$ if $\ell_j = \overline{x}_i$ for $j \in [3]$. Note that $G_\phi$ consists only of edges between $V_i$ and $\overline{V}_i$, for $i \in [r]$. For the proof of correctness, note that for every model $\mathcal{M}$ of $\phi$, either $[\![\Pr(V_i = 1)]\!]_{\mathcal{M}} = 0$ or $[\![\Pr(\overline{V}_i = 1)]\!]_{\mathcal{M}} = 0$ holds, depending on the relative order of $V_i$ and $\overline{V}_i$ in the well-order $\prec$ in $\mathcal{M}$. $\square$

We contrast the hardness obtained in Theorem 5.1 by considering the number of variables in $\mathbf{V}$ as a new parameter. Towards this goal, Lemma 5.2 establishes the existence of a well-structured model for every YES-instance.

**Lemma 5.2.** *Let $\phi$ over domain $D$ be a YES-instance of* SAT$^{\text{poly}}_{\text{counterfact}}$ *over variables $\mathbf{V}$. There is an ordering $V_1, \ldots, V_n$ of $\mathbf{V}$ such that the $\phi$ is satisfied by a model $\mathcal{M} = (\mathbf{V}, \mathbf{U}, \mathcal{F}, \mathbb{P})$ with the following properties: for each $i \in [n]$, let $Q_i$ be the set of all possible functions mapping the values of $V_1, \ldots, V_{i-1}$ to a value of $V_i$, that is, the set of all functions from $D^{i-1}$ to $D$ (with $Q_1$ simply being a set of constant functions). Then*

- $\mathbf{U} = \{U\}$ *with* $\text{Val}(U) = Q_1 \times \ldots \times Q_n$, *where $U[i] \in Q_i$ denotes the $i^{th}$ entry in $U$; and*

- $\mathcal{F} = \{f_{V_i} \mid i \in [n]\}$ *with* $f_{V_i}(U, V_1, \ldots, V_{i-1}) := U[i](V_1, \ldots, V_{i-1})$.

*Proof.* Let $\mathcal{M}' = (\mathbf{V}, \mathbf{U}', \mathcal{F}', \mathbb{P}')$ be any model witnessing that $\phi$ is a YES-instance. Without loss of generality, we assume $\mathbf{U}'$ to consist of a single variable $U'$: If there were multiple variables $U_1, U_2, \ldots, U_\ell$ in $\mathbf{U}'$, we could replace them by some $U'$ with $\text{Val}(U') = \text{Val}(U_1) \times \text{Val}(U_2) \times \ldots \times \text{Val}(U_\ell)$, and update $\mathbb{P}'$ and $\mathcal{F}'$ accordingly.

Consider an ordering $V_1, \ldots, V_n$ of $\mathbf{V}$ respecting the well-order $\prec$ of $\mathcal{M}'$. For each $i \in [n]$, $f'_{V_i} \in \mathcal{F}$ describes $V_i$ as a function of $U'$ and $V_1, \ldots, V_{i-1}$. Partition $\text{Val}(U')$ such that there is a class $C_q$ for each $q = (q_1, \ldots, q_n) \in (Q_1 \times \ldots \times Q_n)$ and let it contain $u \in \text{Val}(U')$ if and only if for all $i \in [n]$ and $\overline{s} \in D^{i-1}$, we have that $f'_{V_i}(u, \overline{s}) = q_i(\overline{s})$. Now each $u \in \text{Val}(U')$ belongs to precisely one class $C_q$.

We construct the model $\mathcal{M} = (\mathbf{V}, \mathbf{U}, \mathcal{F}, \mathbb{P})$ where $\mathbf{U}$ and $\mathcal{F}$ are defined as specified above and $\mathbb{P}$ is such that for each

$q \in \text{Val}(U)$ we have $\mathbb{P}(U = q) = \sum_{u' \in C_q} \mathbb{P}'(U' = u')$ (with $\mathbb{P}(U = q) = 0$ if $C_q = \emptyset$). Note that this yields a well-defined probability distribution over $U$. The model $\mathcal{M}$ satisfies $\phi$ since every term $\Pr(\varepsilon)$ over $\mathbf{V}$ has the same probability in $\mathcal{M}$ and $\mathcal{M}'$. Indeed, for every class $C_q$ and each event $\varepsilon$, we have that $\mathcal{F}', u \models \varepsilon$ either for all $u \in C_q$ or no $u \in C_q$, as by definition all these $u$ result in the exact same values for the variables in $\mathbf{V}$, even under interventions. Furthermore, $\mathcal{F}$ is such that $\mathcal{F}, q \models \varepsilon$ if and only if $\mathcal{F}', u \models \varepsilon$ for all $u \in C_q$. For any event $\varepsilon \in \mathcal{E}_{\text{counterf}}$, recall that $S_{\mathcal{M}}(\varepsilon) \subseteq \text{Val}(U)$ and $S_{\mathcal{M}'}(\varepsilon) \subseteq \text{Val}(U')$ denote the sets of values of hidden variables such that $\varepsilon$ happens in the respective model. We proved that $S_{\mathcal{M}'}(\varepsilon) = \bigcup_{q \in S_{\mathcal{M}}(\varepsilon)} C_q$ and thus, by definition of $\mathbb{P}$, event $\varepsilon$ happens in both models with the same probability, that is, $[\![\Pr(\varepsilon)]\!]_{\mathcal{M}} = [\![\Pr(\varepsilon)]\!]_{\mathcal{M}'}$. Hence, $\mathcal{M}$ witnesses $\phi$ to be a YES-instance as well. $\square$

**Theorem 5.3.** SAT$^{\text{lin}}_{\text{counterfact}}$ *is in* FPT *w.r.t. the combined parameter $d + n$.*

*Proof.* Given an instance $\phi$ over domain $D$, we perform the following for each of the $n!$ orders of variables. If we do not find a model for any of these orders, we reject the instance. Consider a total order $V_1, \ldots, V_n$ of $\mathbf{V}$ and let $U$, $\mathcal{F}$, and the $Q_i$ be defined as in Lemma 5.2. We test whether $\phi$ admits a model as described in Lemma 5.2 with respect to that order using the following LP. Create an LP-variable $p_q$ for each $q \in Q_1 \times \ldots \times Q_n$. These represent the probability distribution over $U$, so we add an LP-constraint $\sum_{q \in Q_1 \times \ldots \times Q_n} p_q = 1$ and, for each such $q$, we add an LP-constraint $p_q \geq 0$. Furthermore, using Observation 3.3, we transform each constraint in $\phi$ into an LP-constraint by replacing each term $\Pr(\varepsilon)$ by a sum over all variables $p_q$ for which $\mathcal{F}, q \models \varepsilon$. We accept the instance $\phi$ if and only if there is at least one ordering of $\mathbf{V}$ for which the constructed LP has a solution. In each LP there are $\mathcal{O}(|\phi| + |Q_1 \times \ldots \times Q_n|)$ LP-constraints and the number of variables is at most

$$|Q_1 \times \ldots \times Q_n| = \prod_{i=1}^{n} |Q_i| \leq \prod_{i=1}^{n} d^{d^{i-1}} \leq d^{d^n}.$$

As we can find a solution to an LP (or decide that there is none) in polynomial time with respect to its size (that is, the number of variables plus constraints), the total running time of the algorithm is $n!(|\phi|^{\mathcal{O}(1)} + d^{\mathcal{O}(d^n)})$.

If one of the constructed LPs has a solution, let $\mathbb{P}$ be the probability distribution over $U$ described by the variables $p_q$ in that solution. It is straight-forward to verify that $(\mathbf{V}, \{U\}, \mathcal{F}, \mathbb{P})$ is a model for $\phi$, so accepting the instance is correct. For the other direction, suppose $\phi$ is a YES-instance. Then there is a well-structured model $\mathcal{M} = (\mathbf{V}, \{U\}, \mathcal{F}, \mathbb{P})$ for some ordering of $\mathbf{V}$ by Lemma 5.2. Observe that the LP constructed for that ordering of the variables has a solution

by setting $p_q := \mathbb{P}(U = q)$ for each $q \in \mathrm{Val}(U)$, so the algorithm indeed accepts the instance. $\qquad\square$

## 6. Concluding Remarks

While previous works have focused on mapping the complexity lower bounds for SATISFIABILITY in Pearl's Causal Hierarchy, the parameterized paradigm allows us to identify islands of tractability for the problem. Our contributions include not only these positive results, but also lower bounds which show that the obtained complexity classifications are tight.

Our work indicates that in applications where only the "shallow" fragments of PCH arise, it may be worthwhile to exploit structural properties of instances before feeding these to a general solving procedure. An analysis of which structural properties occur in distinct practical settings is identified as an interesting research direction. Moreover, our findings open up further avenues for future work:

- **The Impact of Marginalization.** As mentioned in Section 1, recent works (van der Zander et al., 2023; Dörfler et al., 2025) have proposed an enrichment of the expressivity matrix $M$ via the marginalization operator $\sum$. It would be interesting to explore possible extensions of our approaches to this setting—in particular, are these enriched fragments of the PCH tractable without bounding the nesting depth of $\sum$?

- **Treewidth-Guided Linear Programming.** To the best of our knowledge, the proofs of Results (**1**) and (**2**) rely on an entirely novel approach to establishing parameterized tractability with respect to treewidth. Since this approach is tailored to problems that combine discrete structures (graphs) with non-discrete elements (e.g., probabilities), we would not be surprised to see further applications of the technique in the domains of artificial intelligence and machine learning.

- **XP-Algorithm for $\mathrm{SAT}^{\mathrm{lin}}_{\mathrm{counterfact}}$ par. by $n$ alone.** While our results paint an almost complete picture of the problem's parameterized complexity, they leave open the existence of a polynomial-time algorithm for $\mathrm{SAT}^{\mathrm{lin}}_{\mathrm{counterfact}}$ over constantly many variables.

- **Structural Restrictions.** Another promising direction for further research is to consider satisfiability in the PCH with further restrictions on the desired SCM. For example, the "structure" of the model (i.e., variable dependencies) could be provided as an additional input. The classical complexity of this problem version has been studied recently (van der Zander et al., 2023); lifting these novel results to the parameterized realm poses an interesting challenge for future research.

Finally, we note that the very recent work of Yang & Bareinboim (2025) considers a more refined landscape for the PCH by defining intermediate layers between causal and counterfactual. Since all our complexity-theoretic results place the causal and counterfactual layers on equal footing, both the algorithms and lower bounds carry over to these newly introduced layers of the PCH.

## Impact Statement

This paper presents work whose goal is to advance the field of Machine Learning. There are many potential societal consequences of our work, none which we feel must be specifically highlighted here.

## Acknowledgments

This research was funded in whole or in part by the Austrian Science Fund (FWF), projects 10.55776/COE12 and 10.55776/Y1329.

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
