# Gateways to Tractability for Satisfiability in Pearl's Causal Hierarchy (Appendix: Full Version)

## Abstract

Pearl's Causal Hierarchy (PCH) is a central framework for reasoning about probabilistic, interventional, and counterfactual statements, yet the satisfiability problem for PCH formulas is computationally intractable in almost all classical settings. We revisit this challenge through the lens of parameterized complexity and identify the first gateways to tractability. Our results include fixed-parameter and XP-algorithms for satisfiability in key probabilistic and counterfactual fragments, using parameters such as primal treewidth and the number of variables, together with matching hardness results that map the limits of tractability. Technically, we depart from the dynamic programming paradigm typically employed for treewidth-based algorithms and instead exploit structural characterizations of well-formed causal models, providing a new algorithmic toolkit for causal reasoning.

## 1. Introduction

Pearl's Causal Hierarchy (PCH) (Shpitser & Pearl, 2008; Pearl, 2009; Bareinboim et al., 2022) is a central pillar of the modern theory of causality that is employed in artificial intelligence and other reasoning tasks—see, e.g., the recent book (Pearl & Mackenzie, 2018) on the topic. The PCH is a framework that has three basic layers of depth which capture three fundamental degrees of sophistication for analyzing causal effects and relationships. All of these layers provide a means of formalizing statements via formulas capturing the behavior of a set of probabilistic variables in a *Structural Causal Model* (SCM) (Glymour et al., 1987; Pearl, 2009; Koller & Friedman, 2009; Elwert, 2013), which is a well-established representation of systems with observed as well as hidden variables over a specified domain and their mutual

dependencies. As a basic illustrative example, the statement "the likelihood of having both diabetes ($D$ = yes) and blood type B+ ($T$ = B+) is at most 1%" can be expressed by the formula $\psi = \Pr(D = \text{yes} \wedge T = \text{B+}) \leq 0.01$.

The formula $\psi$ above belongs to the first, basic layer of the PCH—that is, the layer $\mathcal{L}_{\text{prob}}$ of probabilistic reasoning that captures direct statements one can make about the probabilities of certain outcomes. The second layer, $\mathcal{L}_{\text{causal}}$, expands on the basic probability terms in $\mathcal{L}_{\text{prob}}$ via the introduction of Pearl's do-operator (Pearl, 2009) which captures interventional causal reasoning. A basic example of an event that can be captured on this layer of the PCH is contracting a disease after being vaccinated against that disease; the probability of this event can be expressed using the term $\Pr([Y = \text{vaccinated}]\ X = \text{contracted})$[1], where the square brackets denote an intervention that is applied before observing the outcome.[2] Hence, the second layer of the PCH allows us to make statements such as $\Pr([Y = \text{vaccinated}]\ X = \text{contracted}) < \Pr(X = \text{contracted})$. The third layer $\mathcal{L}_{\text{counterfact}}$ of the PCH expands on $\mathcal{L}_{\text{causal}}$ by allowing interventions to be chained, and enables complex statements related to counterfactual situations. For instance, a third-layer term such as $\Pr\big(([M = \text{yes}]\ H = \text{no}) \mid (M = \text{no} \wedge H = \text{yes})\big)$ can express the probability that a patient who did not take medication ($M$) and was hospitalized ($H$) would have avoided hospitalization if he had taken the medication. Formal definitions of these as well as related notions are available in Section 2.

While the three layers of depth of the PCH focus on the expressivity inside the probability term $\Pr(\cdot)$, there is a second dimension to the PCH—specifically, the *breadth* of opera-

---

[1]Anonymous Institution, Anonymous City, Anonymous Region, Anonymous Country. Correspondence to: Anonymous Author <anon.email@domain.com>.

Preliminary work. Under review by the International Conference on Machine Learning (ICML). Do not distribute.

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

$$\mathcal{E}_{\text{prop}} ::= V = v \mid \neg\mathcal{E}_{\text{prop}} \mid \mathcal{E}_{\text{prop}} \wedge \mathcal{E}_{\text{prop}},$$
$$\mathcal{E}_{\text{int}} ::= \top \mid V = v \mid \mathcal{E}_{\text{int}} \wedge \mathcal{E}_{\text{int}},$$
$$\mathcal{E}_{\text{post-int}} ::= [\mathcal{E}_{\text{int}}] \, \mathcal{E}_{\text{prop}},$$
$$\mathcal{E}_{\text{counterf}} ::= \mathcal{E}_{\text{post-int}} \mid \neg\mathcal{E}_{\text{counterf}} \mid \mathcal{E}_{\text{counterf}} \wedge \mathcal{E}_{\text{counterf}}.$$

We call events in $\mathcal{E}_{\text{prop}}$ *propositions* and events in $\mathcal{E}_{\text{int}}$ *interventions*. Each event $\varepsilon$ can only occur within a probabilistic statement $\Pr(\varepsilon)$, called a *term*. The *size of a term* is the number of its atoms. For $\mathcal{E} \in \{\mathcal{E}_{\text{prop}}, \mathcal{E}_{\text{post-int}}, \mathcal{E}_{\text{counterf}}\}$ and $\varepsilon \in \mathcal{E}$, we define the following ways of combining terms.

$$T_{\text{base}}(\mathcal{E}) ::= \Pr(\varepsilon),$$
$$T_{\text{lin}}(\mathcal{E}) ::= \Pr(\varepsilon) \mid T_{\text{lin}}(\mathcal{E}) + T_{\text{lin}}(\mathcal{E}),$$
$$T_{\text{poly}}(\mathcal{E}) ::= \Pr(\varepsilon) \mid T_{\text{poly}}(\mathcal{E}) + T_{\text{poly}}(\mathcal{E}) \mid$$
$$T_{\text{poly}}(\mathcal{E}) \cdot T_{\text{poly}}(\mathcal{E}).$$

Last, for $* \in \{\text{base}, \text{lin}, \text{poly}\}$ we define $\mathcal{L}^*_{\text{prob}}$, $\mathcal{L}^*_{\text{causal}}$, and $\mathcal{L}^*_{\text{counterfact}}$ to be the *languages* that contain all sets of inequalities over elements in $T_*(\mathcal{E}_{\text{prop}})$, $T_*(\mathcal{E}_{\text{post-int}})$, and $T_*(\mathcal{E}_{\text{counterf}})$, respectively. The elements inside $\mathcal{L}^*_{\text{prob}}$, $\mathcal{L}^*_{\text{causal}}$, and $\mathcal{L}^*_{\text{

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

**The Class $\exists\mathbb{R}$.** The Existential Theory of the Reals (ETR) contains all true sentences of the form

$$\exists x_1 \ldots \exists x_k \phi(x_1, \ldots, x_k),$$

where $\phi$ is a quantifier-free Boolean formula over the basis $\{\wedge, \vee, \neg\}$ and a signature consisting of constant symbols (0 and 1), function symbols ($+$ and $\cdot$), and predicates ($<, \leq$, and $=$). The sentence is interpreted over the real numbers in the standard way. The closure of ETR under polynomial time many-one reductions yields the complexity class $\exists\mathbb{R}$ (Grigoriev & Jr., 1988; Canny, 1988). For a comprehensive compendium on $\exists\mathbb{R}$, see (Schaefer et al., 2024); here, we only require the class for the lower bound established in Theorem 3.2.

# 3. Satisfiability for Languages of PCH and Structural Insights

We examine several analogues of the well-known problem BOOLEAN SATISFIABILITY that capture various probabilistic, causal, and counterfactual statements. We denote these problems as $\text{SAT}_\circledast^*$, where $\circledast \in \{\text{prob}, \text{causal}, \text{counterfact}\}$ and $* \in \{\text{base}, \text{lin}, \text{poly}\}$, and define them as follows.

---

$\text{SAT}_\circledast^*$
**Input:** A set $D$ of $d$ domain values and a formula $\phi \in \mathcal{L}_\circledast^*$ over variables $\mathbf{V} = \{V_1, \ldots, V_n\}$.
**Task:** Decide if there exists a recursive Structural Causal Model $\mathcal{M}$ over $D$ such that $\mathcal{M} \models \phi$.

---

The classical computational complexity of $\text{SAT}_\circledast^*$ has by now been studied extensively (Fagin et al., 1990; Ibeling & Icard, 2020; van der Zander et al., 2023; Mossé et al., 2024; Dörfler et al., 2025). We remark that our definition of $\text{SAT}_\circledast^*$ slightly deviates from the one established in previous works, in the sense that we restrict our attention to input formulas $\phi$ that are *sets of inequalities* (that is, each inequality forms a constraint) rather than allowing arbitrary Boolean combinations of inequalities. However, this restriction does not affect any of the known complexity-theoretic results, since previous lower-bound proofs did not employ any Boolean combinations beyond sets. However, the situation changes drastically when studying $\text{SAT}_\circledast^*$ from the viewpoint of parameterized complexity, as we show next.

Let $\text{arbSAT}_{\text{prob}}^{\text{base}}$ denote the version of $\text{SAT}_{\text{prob}}^{\text{base}}$ in which $\phi$ is an arbitrary Boolean combination of inequalities over elements in $T_{\text{base}}(\mathcal{E}_{\text{prob}})$. We justify our restriction to $\text{SAT}_\circledast^*$ by the hardness of $\text{arbSAT}_{\text{prob}}^{\text{base}}$ in a very restricted setting, thus dashing any hope to exploit structural properties of $\phi$.

**Theorem 3.1.** $\text{arbSAT}_{\text{prob}}^{\text{base}}$ *is* NP*-complete even if* $G_\phi$ *is edgeless and* $d = 2$.

*Proof.* Fagin et al. (1990) proved that $\mathrm{arbSAT}^{\mathsf{base}}_{\mathsf{prob}}$ is NP-complete. In order to prove the NP-hardness of $\mathrm{arbSAT}^{\mathsf{base}}_{\mathsf{prob}}$ even for instances in which each constraint consists of only one variable and all variables have domain $D = \{0,1\}$, we perform a reduction from 3-SAT. Let $\Phi := \bigwedge_i C_i$ with $C_i := \bigvee_{j \in [3]} \ell_{i_j}$ be a 3-SAT formula over variables $\mathcal{V}$. Define an instance $\phi$ of $\mathrm{arbSAT}^{\mathsf{base}}_{\mathsf{prob}}$ with $\mathbf{V} = \{V_v \mid v \in \mathcal{V}\}$ over domain $\{0,1\}$ as

$$\phi := \bigwedge_i \Big( \bigvee_{j \in [3]} \Pr(g(\ell_{i_j})) = 1 \Big) \wedge$$
$$\bigwedge_{v \in \mathcal{V}} (\Pr(V_v = 0) = 1 \vee \Pr(V_v = 1) = 1),$$

where $g(\ell_{i_j})$ is replaced by $V_v = 1$ if $\ell_{i_j} = v$, and by $V_v = 0$ if $\ell_{i_j} = \neg v$.

We now argue that $\Phi$ is satisfied if and only if $\phi$ admits an SCM. For the first direction, suppose there exists an assignment $\alpha : \mathcal{V} \to \{0,1\}^{|\mathcal{V}|}$ under which $\Phi$ is satisfied. We construct a model $\mathcal{M} = (\mathbf{V}, \emptyset, \mathcal{F}, \emptyset)$ for $\mathrm{arbSAT}^{\mathsf{base}}_{\mathsf{prob}}$ as follows. For each $v \in \mathcal{V}$, define $f_{V_v} := \alpha(v)$ as a constant function. To see that $\mathcal{M}$ satisfies all constraints in $\phi$, recall that $\alpha$ satisfies at least one literal $\ell_{i_j}$ in each clause $C_i \in \Phi$, that is, $\alpha(v) = 1$ if $\ell_{i_j} = v$, and $\alpha(v) = 0$ if $\ell_{i_j} = \neg v$. The reduction ensures that the $i^{\text{th}}$ conjunct in $\phi$ contains the disjunct $\Pr(V_v = 1) = 1$ in the first, and $\Pr(V_v = 0) = 1$ in the latter case. Since $[\![\Pr(V_v = \alpha(v))]\!]_{\mathcal{M}} = 1$, this satisfies $\phi$.

For the other direction, suppose there exists a model $\mathcal{M}$ satisfying $\phi$. Therefore, either $[\![\Pr(V_v = 0)]\!]_{\mathcal{M}} = 1$ or $[\![\Pr(V_v = 1)]\!]_{\mathcal{M}} = 1$ for each $v \in \mathcal{V}$. We obtain an assignment $\alpha : \mathcal{V} \to \{0,1\}^{|\mathcal{V}|}$ by defining $\alpha(v) = 0$ if and only if $[\![\Pr(V_v = 0)]\!]_{\mathcal{M}} = 1$ for each variable $v \in \mathcal{V}$. Since all conjuncts of $\phi$ are satisfied by $\mathcal{M}$, it holds by construction that $\alpha$ satisfies all clauses of $\Phi$. $\square$

**Intractability of $\mathrm{SAT}^{\mathsf{poly}}_{\mathsf{prob}}$.** Our main contributions target the lin and base fragments of the expressivity matrix. Here, we show that the tractability results obtained in Sections 4 and 5 most likely not hold for polynomial inequalities.

**Theorem 3.2.** $\mathrm{SAT}^{\mathsf{poly}}_{\mathsf{prob}}$ is $\exists\mathbb{R}$-*complete even if* $n = 1$, *or if* $d = 2$ *and* $G_\phi$ *is edgeless.*

*Proof.* Mossé et al. (2024) proved that $\mathrm{SAT}^{\mathsf{poly}}_{\mathsf{prob}}$ is $\exists\mathbb{R}$-complete. Using a different reduction, we first prove our second statement by showing hardness on instances in which no two variables co-occur in a term and all variables have a binary domain. Our proof conceptually resembles the construction used by van der Zander et al. (2023, Proposition 6.5). Consider the following problem.

---

$\mathrm{ETR}^{1/8,+,\times}_{[-1/8,1/8]}$

**Input:** A set $S$ of equations over real variables $x_1, \ldots, x_n \in \left[-\frac{1}{8}, \frac{1}{8}\right]$ in which each equation is of the form $x_i = \frac{1}{8}$ or $x_{i_1} + x_{i_2} = x_{i_3}$ or $x_{i_1} x_{i_2} = x_{i_3}$, and $i, i_1, i_2, i_3 \in [n]$.

**Task:** Decide if $S$ has a solution.

---

Note that $\mathrm{ETR}^{1/8,+,\times}_{[-1/8,1/8]}$ is known to be $\exists\mathbb{R}$-complete (Abrahamsen et al., 2022). We show that every instance of $\mathrm{ETR}^{1/8,+,\times}_{[-1/8,1/8]}$ can be reduced to an instance of $\mathrm{SAT}^{\mathsf{poly}}_{\mathsf{prob}}$, in which all variables have a binary domain and each term speaks about just one variable.

Let $S$ be an instance of $\mathrm{ETR}^{1/8,+,\times}_{[-1/8,1/8]}$ that contains variables $x_1, \ldots, x_n$ and let $V_1, \ldots, V_n$ be binary random variables. We obtain an instance $\phi$ over domain $D = \{0,1\}$ of $\mathrm{SAT}^{\mathsf{poly}}_{\mathsf{prob}}$ in time $O(|\phi|)$ by replacing for all $i \in [n]$ each occurrence of $x_i$ in $S$ by the expression $e_i := \frac{1}{4}\Pr(V_i = 0) - \frac{1}{8}$. Note that in $\phi$, no two random variables co-occur in the same term and $d = 2$. We now argue that $S$ is satisfiable if and only if $\phi$ has a model. First, suppose there exists a solution for $S$, i.e., a function $f$ that maps each variable $x_i, i \in [n]$ to a value in $\left[-\frac{1}{8}, \frac{1}{8}\right]$ such that all equations are satisfied under $f$. We obtain a model $\mathcal{M}$ for $\phi$ by introducing a hidden variable $U_i$ for each binary random variable $V_i$ in $\phi$ and define $f_{V_i}$ such that $V_i = U_i$. Then, letting $\mathbb{P}(U_i = 0) := 4f(x_i) + \frac{1}{2}$ for all $i \in [n]$ will satisfy $\phi$, since it enforces $0 \leq [\![\Pr(V_i = 0)]\!]_{\mathcal{M}} \leq 1$ and ensures that $e_i = f(x_i)$. Likewise, if there exists a model $\mathcal{M}$ for $\phi$, then we can determine $[\![\Pr(V_i = 0)]\!]_{\mathcal{M}}$ and derive $e_i$. Setting $x_i$ to the value $e_i$ for all $i \in [n]$ thus solves all equations in $S$ and ensures $x_i \in \left[-\frac{1}{8}, \frac{1}{8}\right]$.

The above construction is easily adapted to show hardness even if $n = 1$. To construct $\phi$, let $\mathbf{V} = \{V\}$ with $d = n + 1$. For each $i \in [n]$, add a constraint $\Pr(V = i) \leq \frac{1}{n}$. Furthermore, add a constraint for each constraint in $S$ that, for $i \in [n]$, replaces each occurrence of $x_i$ by $e_i := \frac{n}{4}\Pr(V = i) - \frac{1}{8}$, which scales $\Pr(V = i)$ to be in $\left[-\frac{1}{8}, \frac{1}{8}\right]$. This construction holds by the same arguments as employed above, where the value 0 in the range of $V$ serves as a buffer so that for $i \in [n]$ the probability $\Pr(V = i)$ can take arbitrary values in $\left[0, \frac{1}{n}\right]$. $\square$

Despite the hardness of sets of polynomial inequalities even in the absence of interventions, we remark that one can still obtain exponential time algorithms by employing the constructions in Theorems 4.1 and 5.3. Using the same approaches now requires solving systems of polynomial inequalities instead of LPs. These $\exists\mathbb{R}$ instances can be solved, for example, by invoking Renegar's Theorem (Renegar, 1992a;b;c).

**Further Structural Insights in $\mathrm{SAT}^*_\circledast$.** In order to facilitate our complexity-theoretic analysis, we emphasize that a Structural Causal Model can be efficiently evaluated, that is, given the values of its hidden variables, it can be decided in polynomial time, whether a certain event happens.

**Observation 3.3.** *Given a model $\mathcal{M} = (\mathbf{U}, \mathbf{V}, \mathcal{F}, \mathbb{P})$, an event $\varepsilon \in \mathcal{E}_{\mathrm{counterf}}$, and some $\overline{u} \in \mathrm{Val}(\mathbf{U})$, let $|\varepsilon|$ denote the number of atoms in $\varepsilon$. Assuming that each function in $\mathcal{F}$ can be evaluated in time $\mathcal{O}(n)$, one can decide whether $\mathcal{F}, \overline{u} \models \varepsilon$ in time in $\mathcal{O}(n^2 + |\varepsilon|)$.*

*Proof.* The only randomness in a model stems from the hidden variables $\mathbf{U}$. Fixing their values thus deterministically settles whether $\varepsilon$ holds true or not. To decide which one is the case, it suffices to show how to evaluate events from $\mathcal{E}_{\mathrm{post\text{-}int}}$, as $\mathcal{E}_{\mathrm{counterf}}$ is simply a Boolean formula over these events that—having evaluated each event of type $\mathcal{E}_{\mathrm{post\text{-}int}}$—can be evaluated in time in $\mathcal{O}(|\varepsilon|)$. To evaluate an event $[\gamma]\,\varepsilon' \in \mathcal{E}_{\mathrm{post\text{-}int}}$, we compute the value of each variable $V \in \mathbf{V}$ following the implicit well-order $\prec$ of the model. Note that the value of $V$ is either fixed by the intervention $\gamma$ or can be computed from $\overline{u}$ and the values of $\mathbf{V}_{\prec V}$. Once the values of all variables in $\mathbf{V}$ are determined, the probabilistic event $\varepsilon'$ can be evaluated in time in $\mathcal{O}(|\varepsilon'|)$. $\qquad\square$

The running time of our algorithmic results often depends on the size $d$ of the domain $D$. Assuming that $d$ is not much larger than the size of $\phi$ does not reduce the generality of our results, as we can always reduce to an equivalent instance where $d$ is bounded from above by $|\phi| + 1$.

**Observation 3.4.** *Consider an instance of $\mathrm{SAT}^*_\circledast$ consisting of a domain $D$ and a formula $\phi \in \mathcal{L}^*_\circledast$. Let $D_\phi$ be the set of values in $D$ that are explicitly mentioned in at least one atom in $\phi$ and choose some $\gamma \notin D_\phi$. Then, it holds that $\phi$ over domain $D_\phi \cup \{\gamma\}$ is a YES-instance of $\mathrm{SAT}^*_\circledast$ if and only if so is $\phi$ over $D$.*

*Proof.* To show this equivalence, intuitively, we contract all values in $D \setminus D_\phi$ into $\gamma$.

Suppose there is a model $\mathcal{M}$ solving $(\phi, D_\phi \cup \{\gamma\})$. As $\phi$ does not differentiate between values in $\{\gamma\} \cup D \setminus D_\phi$, the model $\mathcal{M}$ also witnesses $(\phi, d)$ to be a YES-instance.

For the other direction, suppose there is a model $\mathcal{M} = (\mathbf{V}, \mathbf{U}, \mathcal{F}, \mathbb{P})$ solving $(\phi, D)$ with an implicit well-order $\prec$, and let $V_1, V_2, \ldots$ denote $\mathbf{V}$ as ordered by $\prec$. Note that, without loss of generality, for each $i \in [n]$ we can assume $f_{V_i} \in \mathcal{F}$ to be represented by a case distinction over the values of $\mathbf{U}$ and $\mathbf{V}_{\prec V}$, where the result of each case is stated as an element in $D$. Exhaustively repeat the following. Let $i$ be minimal such that there is some function $f_V \in \mathcal{F}$ which has a condition $V_i = v$ in the case distinction with $v \notin D_\phi$. Replace the condition by $\underline{f_{V_i}} = v$, where $\underline{f_{V_i}}$ denotes the

term used to compute $f_{V_i}$. Rewrite the updated $f_V$ to once more be a case distinction over the values of $\mathbf{U}$ and $\mathbf{V}_{\prec V}$. Note that this preserves the probability distribution over $f_V$, even under interventions: For every fixed $\overline{u} \in \mathbf{U}$, the variables $V_1, \ldots, V_{i-1}$ are computed the same way as before. Further, by assumption there is no intervention with an atom setting $V_i$ to $v$. Hence, the only way that $V_i = v$ happens is if $\underline{f_{V_i}} = v$.

If at some point there is no $i$ satisfying the condition, we have an alternative but equivalent representation of the functions in $\mathcal{F}$ which do not compare any variable $V \in \mathbf{V}$ to any value outside $D_\phi$. At this point, update the functions once more such that whenever a function $f_V$ would output some value in $D \setminus D_\phi$, it now outputs $\gamma$. As the functions computing the other variables do not compare such a variable $V$ to a value outside of $D_\phi$, all variables are computed in the exact same way as before except that for each variable $V$ all outputs in $D \setminus D_\phi$ are contracted into $\gamma$. This yields an updated model $\mathcal{M}'$ where the domain is restricted to $D_\phi \cup \{\gamma\}$. This model satisfies $\phi$ if so does $\mathcal{M}$. $\qquad\square$

# 4. Linear Inequalities over Probabilistic Expressions

This section is dedicated to the complexity-theoretic analysis of $\mathrm{SAT}^*_{\mathrm{prob}}$, that is, the satisfiability problem for the layer of the PCH that does not allow any interventions. First, we establish the main tractability result of this section, and then proceed by showing that it is tight as outlined in Figure 1.

**Theorem 4.1.** $\mathrm{SAT}^{\mathrm{lin}}_{\mathrm{prob}}$ *is in* FPT *w.r.t. the combined parameter $d + \mathrm{tw}(\phi)$, and in* XP *w.r.t. $\mathrm{tw}(\phi)$.*

*Proof.* Consider an instance of $\mathrm{SAT}^{\mathrm{lin}}_{\mathrm{prob}}$ with formula $\phi$ and domain $D$. We prove both statements simultaneously by describing an algorithm that runs in time $d^{f(\mathrm{tw}(\phi))}|\phi|^{\mathcal{O}(1)}$, for a computable function $f$. Consider a nice tree decomposition of $G_\phi$ consisting of $\mathcal{O}(n)$ nodes with maximum size $w := \mathrm{tw}(\phi) + 1$ computed by, e.g., the algorithm of (Bodlaender, 1996). Without loss of generality, assume that only the bags of leaf nodes are empty and ignore them in the following procedure (instead we consider their parents as leaves). For the remaining tree decomposition $\mathbf{T}$, let $D^{|B|}$ be the combined domain of the variables of bag $B$ in $\mathbf{T}$. We construct the following Linear Program (LP). For each distinct bag $B$ and $\overline{v} \in D^{|B|}$, construct an LP-variable $p_{B=\overline{v}}$ (note that distinct nodes might share a bag). This will capture the probability of the event $B = \overline{v}$, that is, each variable in $B$ takes the respective value in $\overline{v}$. To ensure a valid probability distribution over the LP-variables in each bag $B$, add the LP-constraints

$$p_{B=\overline{v}} \geq 0 \quad \text{for each LP-variable } p_{B=\overline{v}}, \quad \text{and}$$
$$\sum\nolimits_{\overline{v} \in D^{|B|}} p_{B=\overline{v}} = 1 \quad \text{for each bag } B.$$

For every pair of distinct bags $B, B'$ whose nodes are adjacent in $\mathbf{T}$ and $B \neq B'$, note that there is some $V \in \mathbf{V}$ such that, without loss of generality, $B' = B \cup \{V\}$. To guarantee consistency between the probability distributions of $B$ and $B'$, we add for each such pair and each $\overline{v} \in D^{|B|}$ the LP-constraint

$$p_{B=\overline{v}} = \text{sum}\{p_{B'=\overline{v}'} \mid \overline{v}' \in D^{|B'|} \text{ and } \overline{v}' \text{ sets } B \text{ to } \overline{

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

*Example* 4.2 (Construction in Theorem 4.1). Consider the following instance $\phi$ of $\text{SAT}_{\text{base}}^{\text{lin}}$ with endogenous variables $\mathbf{V} = \{V_1, V_2, V_3, V_4\}$ over domain $D = \{0, 1\}$.

$$\Pr(V_1 = 1 \wedge V_3 = 1) \geq \tfrac{1}{2}$$
$$\Pr(V_2 = 1 \vee V_3 = 1) - 2\Pr(V_3 = 1 \vee V_4 = 1) \geq 0$$
$$\Pr(V_4 = 1) \geq \tfrac{1}{3}$$

The corresponding primal graph $G_\phi$ and a nice tree decomposition of $G_\phi$ are as follows.

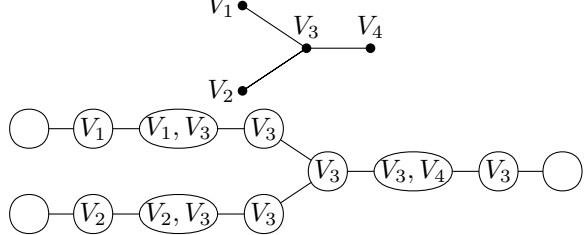

We construct a linear program as follows. For each non-empty bag, we create one LP-variable for each possible assignment of the variables in the bag. This yields variables $p_{V_i=0}$ and $p_{V_i=1}$ for $i \in [3]$ as well as $p_{V_1=x,V_3=y}$ and $p_{V_2=x,V_3=y}$ and $p_{V_3=x,V_4=y}$ for all pairs $x, y \in \{0, 1\}$. Note that for clarity we here write $p_{V_i=x,V_j=y}$ instead

of $p_{B=(x,y)}$ with $B = \{V_i, V_j\}, i < j$. We introduce the following LP-constraints for $i \in [3], (a,b) \in \{(1,3),(2,3),(3,4)\}, x,y \in \{0,1\}$:

$$p_{V_i=0} \geq 0, \quad p_{V_i=1} \geq 0, \quad p_{V_i=0} + p_{V_i=1} = 1,$$
$$p_{V_a=x, V_b=y} \geq 0,$$
$$p_{V_a=0,V_b=0} + p_{V_a=0,V_b=1} + p_{V_a=1,V_b=0}$$
$$+ p_{V_a=1,V_b=1} = 1.$$

To ensure consistency between bags, add the following constraints. For $(a,b) \in \{(1,3),(2,3),(3,4)\}, x \in \{0,1\}$ add

$$p_{V_a=x} = p_{V_a=x,V_b=0} + p_{V_a=x,V_b=1}$$

and for $(a,b) \in \{(1,3),(2,3)\}, x \in \{0,1\}$ add

$$p_{V_b=x} = p_{V_a=0,V_b=x} + p_{V_a=1,V_b=x}.$$

Last, the following three LP-constraints encode the constraints in $\phi$:

$$p_{V_1=1,V_3=1} \geq \tfrac{1}{2}$$
$$p_{V_2=1,V_3=0} + p_{V_2=1,V_3=1} + p_{V_2=0,V_3=1} -$$
$$2(p_{V_3=1,V_4=0} + p_{V_3=1,V_4=1} + p_{V_3=0,V_4=1}) \geq 0$$
$$p_{V_3=0,V_4=1} + p_{V_3=1,V_4=1} \geq \tfrac{1}{3}.$$

Here, $\phi$ is a YES-instance of $\mathrm{SAT}_{\mathrm{base}}^{\mathrm{lin}}$ and satisfied by an SCM $\mathcal{M} = (\mathbf{V}, \{U\}, \mathcal{F}, \mathbb{P})$ such that $\mathbb{P}(U = 1) = \tfrac{1}{2}$ holds[3] as well as

$$f_{V_1}(U) := U, \quad f_{V_2} := 1,$$
$$f_{V_3}(V_1) := V_1, \quad f_{V_4}(V_1) := V_1,$$

which corresponds to the LP-solution where all LP-variables have value 0 except, for $i \in \{1,3\}$,

$$p_{V_2=1} = 1,$$
$$p_{V_i=0} = p_{V_i=1} = p_{V_1=0,V_3=0} =$$
$$p_{V_1=1,V_3=1} = p_{V_2=1,V_3=0} = p_{V_2=1,V_3=1} =$$
$$p_{V_3=0,V_4=0} = p_{V_3=1,V_4=1} = \tfrac{1}{2}.$$

Likewise, this solution to the LP yields a model $\mathcal{M}' = (\mathbf{V}, \mathbf{U}'\mathcal{F}', \mathbb{P}')$ witnessing $\phi$ to be a YES-instance, as constructed by passing through the tree decomposition in order $\{V_1\}, \{V_1, V_3\}, \{V_3\}, \{V_3, V_4\}, \{V_2, V_3\}, \{V_2\}$. We illustrate the first steps towards constructing $\mathcal{M}'$. First, we introduce a hidden variable $U_{V_1}$ with $\mathbb{P}'(U_{V_1} = 1) = \tfrac{1}{2}$ and let $f'_{V_1}(U_{V_1}) := U_{V_1}$. Next we construct hidden variables $U_{V_3|V_1=0}$ and $U_{V_3|V_1=1}$, where $\mathbb{P}'(U_{V_3|V_1=0} = 1) = 0$ and $\mathbb{P}'(U_{V_3|V_1=1} = 1) = 1$, and define

$$f_{V_3}(V_1, U_{V_3|V_1=0}, U_{V_3|V_1=1}) := \begin{cases} U_{V_3|V_1=0}, & \text{if } V_1 = 0; \\ U_{V_3|V_1=1}, & \text{if } V_1 = 1. \end{cases}$$

---

[3]Here, for binary variables we just state the probability of one case; the probability for the other immediately follows.

Defining the remaining observed variables analogously yields an SCM $\mathcal{M}'$ which satisfies $\phi$.

We now show that under well-established complexity assumptions, parameterization by $d$ alone cannot yield tractability, even if the primal graph $G_\phi$ has bounded degree.

**Theorem 4.3.** $\mathrm{SAT}_{\mathrm{prob}}^{\mathrm{base}}$ *is NP-complete even if $d = 2$ and the maximum degree of $G_\phi$ is 8.*

*Proof.* The containment in NP follows from $\mathrm{arbSAT}_{\mathrm{prob}}^{\mathrm{base}} \in$ NP (Fagin et al., 1990). To show hardness in our restricted setting, we perform a reduction from 3-SAT. Note that 3-SAT remains NP-hard when restricted to formulas in which each variable occurs exactly twice negated and twice non-negated (Darmann & Döcker, 2021). Thus, w.l.o.g., we assume that our formula $\Phi := \bigwedge_i C_i$ with $C_i := \bigvee_{j \in [3]} \ell_{i_j}$ over variables $\mathcal{V}$ has this property. We construct an instance $\phi$ of $\mathrm{SAT}_{\mathrm{prob}}^{\mathrm{base}}$ with $\mathbf{V} = \{V_v \mid v \in \mathcal{V}\}$ and $D = \{0,1\}$ such that for each $C_i \in \Phi$, the constraint $\Pr(\bigvee_{j \in [3]} g(\ell_{i_j})) = 1$ is added to $\phi$, where $g(\ell_{i_j})$ is replaced by $V_v = 1$ if $\ell_{i_j} = v$, and by $V_v = 0$ if $\ell_{i_j} = \neg v$. Since each variable occurs in at most 4 clauses, there are at most 8 other variables it co-occurs with; consequently, $G_\phi$ has a maximum degree of 8. We now argue that $\Phi$ is satisfied if and only if $\phi$ admits an SCM.

Suppose there exists an assignment $\alpha : \mathcal{V} \to \{0,1\}^{|\mathcal{V}|}$ that satisfies $\Phi$. We construct a model $\mathcal{M}$ satisfying $\phi$ as in the proof of Theorem 3.1. The proof that $\mathcal{M}$ satisfies all constraints in $\phi$ is analogous.

For the other direction, suppose there exists a model $\mathcal{M} = (\mathbf{V}, \mathbf{U}, \mathcal{F}, \mathbb{P})$ satisfying $\phi$. From the existence of $\mathbb{P}$, we conclude that there exists an assignment $\alpha_{\mathbf{U}}$ to $\mathbf{U}$ that has a non-zero probability. Moreover, by construction, fixing such an $\alpha_{\mathbf{U}}$ yields a full assignment $\alpha_{\mathbf{V}}$ to $\mathbf{V}$ of non-zero probability. We obtain an assignment $\alpha_\Phi : \mathcal{V} \to \{0,1\}^{|\mathcal{V}|}$ by enforcing $\alpha_\Phi(v) = 0 \Leftrightarrow \alpha_{\mathbf{V}}(V_v) = 0$ for each variable $v \in \mathcal{V}$. We claim that $\alpha_\Phi$ satisfies all clauses in $\Phi$. Suppose the contrary, that is, there exists a clause $C_i = \bigvee_{j \in [3]} \ell_{i_j}$ in $\Phi$ that is not satisfied by $\alpha_\Phi$. Then $\bigvee_{j \in [3]} g(\ell_{i_j})$ is not satisfied by $\alpha_{\mathbf{V}}$. However, since $\alpha_{\mathbf{V}}$ has non-zero probability, it follows that $[\![\Pr(\bigvee_{j \in [3]} g(\ell_{i_j}))]\!]_{\mathcal{M}} < 1$, thus, $\mathcal{M}$ fails to satisfy all constraints in $\phi$ which contradicts the fact that it is a model. We can therefore conclude that $\alpha_\Phi$ is a satisfying assignment for $\Phi$. □

The following result complements Theorem 4.3 by ruling out fixed-parameter tractable algorithms for $\mathrm{SAT}_{\mathrm{prob}}^{\mathrm{base}}$ under a different parameterization, namely the number $n$ of variables. Note that since $\mathrm{tw}(\phi) \leq n$, this implies that we should not expect the primal treewidth of a graph to yield fixed-parameter tractability for $\mathrm{SAT}_{\mathrm{prob}}^{\mathrm{base}}$ alone.

**Theorem 4.4.** $\mathrm{SAT}_{\mathrm{prob}}^{\mathrm{base}}$ *is W[1]-hard w.r.t. $n$.*

*Proof.* We reduce from MULTICOLORED-CLIQUE, which asks, given a properly vertex-colored graph $G$ with colors $1, \ldots, k$ and vertices $v_1, \ldots, v_r$, whether $G$ contains a $k$-clique. Given $G$, we construct an instance $\phi$ of $\text{SAT}_{\text{prob}}^{\text{base}}$ as follows. Let $\mathbf{V} = \{V_1, \ldots, V_k\}$ and $D = \{v_1, \ldots, v_r\}$. For each $i \in [k]$ and $a \in [r]$, add the constraint $\Pr(V_i = v_a) \leq 0$, unless $v_a$ has color $i$. For each non-adjacent $v_a, v_b$ with $a < b$ and colors $i, j$, add the constraint $\Pr(V_i = v_a \wedge V_j = v_b) \leq 0$. The construction takes polynomial time and sets $n = k$.

Suppose $G$ contains a clique of size $k$. Let $\alpha \colon [k] \to \{v_1, \ldots, v_r\}$ be such that for every $i \in [k]$ the clique contains vertex $\alpha(i)$ of color $i$. Consider the model $(\mathbf{V}, \emptyset, \mathcal{F}, \emptyset)$, where $\mathcal{F}$ is such that $f_{V_i} \coloneqq \alpha(i)$ is a constant function for each $V_i \in \mathbf{V}$. Clearly, this satisfies each constraint of the form $\Pr(V_i = v_a) \leq 0$. Assume this would not satisfy a constraint of the form $\Pr(V_i = v_a \wedge V_j = v_b) \leq 0$. Then $\alpha(i) = v_a$ and $\alpha(j) = v_b$, so both $v_a$ and $v_b$ are in the clique and thereby adjacent, which contradicts the existence of the constraint.

For the other direction, assume there is a model $\mathcal{M}$ satisfying the $\text{SAT}_{\text{prob}}^{\text{base}}$ instance. Then there is at least one assignment $\overline{v} \in D^k$ such that $[\![\Pr(\mathbf{V} = \overline{v})]\!]_{\mathcal{M}} > 0$. Let $\alpha \colon [k] \to \{v_1, \ldots, v_r\}$ be such that for each $i \in [k]$ we have $V_i = \alpha(i)$ in this assignment. We argue that the vertices $\alpha(1), \ldots, \alpha(k)$ form a clique. Towards a contradiction, assume there are $i, j \in [k], i \neq j$ such that $\alpha(i)$ and $\alpha(j)$ are non-adjacent. Then there is a constraint $\Pr(V_i = \alpha(i) \wedge V_j = \alpha(j)) \leq 0$, which contradicts the model being a solution to the instance as $[\![\Pr(V_i = \alpha(i) \wedge V_j = \alpha(j))]\!]_{\mathcal{M}} \geq [\![\Pr(\mathbf{V} = \overline{v})]\!]_{\mathcal{M}} > 0$. $\square$

## 5. Linear Inequalities over Causal or Counterfactual Expressions

In this section, we turn our attention to interventional causal reasoning. We initiate our study by showing that the FPT-tractability that was established in Theorem 4.1 does not carry over.

**Theorem 5.1.** $\text{SAT}_{\text{causal}}^{\text{lin}}$ *is NP-complete even if $d = 2$ and all edges in $G_\phi$ are pairwise vertex-disjoint.*

*Proof.* Fagin et al. (1990) showed that $\text{arbSAT}_{\text{causal}}^{\text{lin}}$ is NP-complete. To show NP-hardness even under the stated restrictions, we reduce from 3-SAT. Consider a 3-SAT formula $\Phi$ with $r$ variables. We define an instance $\phi$ of $\text{SAT}_{\text{causal}}^{\text{lin}}$ with domain $D = \{0, 1\}$ as follows. For each variable $x_i \in \Phi$, we introduce endogenous variables $V_i$ and $\overline{V}_i$, as well as the constraints

$$\Pr([V_i = 1] \, \overline{V}_i = 1) = 0, \text{ and } \Pr([\overline{V}_i = 1] \, V_i = 1) = 0.$$

Furthermore, for each clause $\ell_1 \vee \ell_2 \vee \ell_3$ in $\Phi$, we add $\Pr(L_1 = 1) + \Pr(L_2 = 1) + \Pr(L_3 = 1) \geq 1$ to $\phi$, where, $L_j = V_i$ if $\ell_j = x_i$, and $L_j = \overline{V}_i$ if $\ell_j = \overline{x}_i$ for $j \in [3]$. Note that $G_\phi$ consists only of edges between $V_i$ and $\overline{V}_i$, for $i \in [r]$. We now argue that $\Phi$ is satisfiable if and only if there is a model for $\phi$. Suppose there exists an assignment $\alpha \colon \text{Var}(\Phi) \to \{0, 1\}^r$ under which $\Phi$ is satisfied. We construct a model $\mathcal{M} = (\mathbf{V}, \emptyset, \mathcal{F}, \emptyset)$ for $\phi$ as follows: First, note that by construction, $\mathbf{V} = \{V_i, \overline{V}_i \mid i \in [r]\}$. For each $i \in [r]$, if $\alpha(x_i) = 1$ then let $\mathcal{F}$ be such that $f_{\overline{V}_i} \coloneqq 0$, which satisfies $\Pr([V_i = 1] \, \overline{V}_i = 1) = 0$, and $f_{V_i} \coloneqq 1 - \overline{V}_i$, which satisfies $\Pr([\overline{V}_i = 1] \, V_i = 1) = 0$. If instead $\alpha(x_i) = 0$, let $f_{V_i} \coloneqq 0$ and $f_{\overline{V}_i} \coloneqq 1 - V_i$, which analogously satisfies both constraints. This yields an SCM $\mathcal{M}$. It remains to show that each clause-constraint is satisfied. Consider a clause $\ell_1 \vee \ell_2 \vee \ell_3$ and let, without loss of generality, $\ell_1$ be TRUE in $\alpha$. Assume that $\ell_1 = x_i$ for some $i \in [r]$ (the case of $\ell_1 = \overline{x}_i$ holds analogously). Then, in the model without interventions, it holds that $\overline{V}_i = 0$ and $V_i = 1 - 0 = 1$ with probability 1, in other words, $[\![\Pr(V_i = 1)]\!]_{\mathcal{M}} = 1$. As $\Pr(V_i = 1)$ is one of the summands in the constraint for clause $i$ and the other summands are non-negative, this satisfies the clause constraint.

For the other direction, suppose there exists a model $\mathcal{M} = (\mathbf{V}, \mathbf{U}, \mathcal{F}, \mathbb{P})$ for $\phi$ with an associated well-order $\prec$ over $\mathbf{V}$. Consider the assignment $\alpha$ obtained by letting $x_i = 1$ if $\overline{V}_i \prec V_i$ and $x_i = 0$, otherwise. Note that if $\overline{V}_i \prec V_i$ then $f_{\overline{V}_i}$ does not depend on $V_i$ and thus the constraint $\Pr([V_i = 1] \, \overline{V}_i = 1) = 0$ implies $[\![\Pr(\overline{V}_i = 1)]\!]_{\mathcal{M}} = 0$. Vice versa, if $V_i \prec \overline{V}_i$, we have $[\![\Pr(V_i = 1)]\!]_{\mathcal{M}} = 0$. As for each clause $\ell_1 \vee \ell_2 \vee \ell_3$ we have that $\Pr(L_1 = 1) + \Pr(L_2 = 1) + \Pr(L_3 = 1) \geq 1$, there is $j \in [3]$ such that $L_j$ does *not* precede its counterpart and thus $\ell_i$ is set to TRUE by $\alpha$. $\