# OpenReview forum: "Gateways to Tractability for Satisfiability in Pearl’s Causal Hierarchy"
_ICML.cc/2026/Conference — ICML 2026 regular_

### Official Review · Reviewer_jmob · 2026-03-09

**Soundness:** 3
**Presentation:** 4
**Significance:** 3
**Originality:** 3
**Overall Recommendation:** 4
**Confidence:** 3

**Summary:**

The paper provides new results in the study of SATISFIABILITY problems defined for structural causal formulae - i.e., for different classes of formulae (Layer 1, 2, 3) and different polynomial degrees, we check whether there is an SCM that permits the formula. On the positive side, the authors show that linear Layer 1 SATISFIABILITY is XP-tractable in terms of the problem's "primal tree width" and FP-tractable in terms of tree width and domain size of the variables involved. On the impossibility side, they show hardness when removing any of the parameters or trying to move to polynomial formulae. The proof technique they highlight does not use DP, but rather uses a linear programming formulation.

**Compliance With Llm Reviewing Policy:**

Affirmed.

**Final Justification:**

I find the work rigorous and interesting. My main objection is the relevance of these results - this has not changed since my initial review. Some aspects of the contributions need to be better motivated, particularly w.r.t the polynomial constraint satisfaction. The motivating examples given to reviewer YKHJ were (per my understanding):
1. An example where a set of linear equalities is being rewritten as square distance loss.
2. Front-door admissible constraints w.r.t a graph
3. Markov-factorization constraints w.r.t a graph.

(1) doesn't need polynomial rewriting. (2) and (3) are multi-linear constraints w.r.t observable variables, so yes, technically “polynomial”. But so far no examples of where I actually care about higher-powers of probability terms involving just observable variables (because PCH probability expressions don't involve noise variables).

And even for (2) and (3) - the constraints are w.r.t a causal DAG. For instance, (2) is equivalent to asking: does the graph satisfy the "front-door" criteria? This is easy to check graphically (in polynomial-time, perhaps using BFS) for a given DAG and a given adjustment set. So what's the point of specifying this as an SCM satisfiability problem?

But I do appreciate the rigor of the work. I retain a weak accept for this reason.

**Key Questions For Authors:**

The motivations for some of these results could be strengthened much further. For instance,

1. What is the motivation for the polynomial class of SAT problems you study? Why would one be looking for SCMs that permit constraints over higher powers of probability terms - it seems an esoteric question given problems generally studied in causal inference.
2. All the SAT problems studied here search over the space of ALL possible SCMs involving these variables, and some of the proof techniques seem to leverage the fact that one can use a single U that determines functional mappings for all observable variables. Why is this a more natural question to ask rather than what is standard in the causal inference literature - limiting the search space to SCMs compatible with a causal DAG given as input to the problem? It's unlikely that researchers are working in environments where a maximally confounded SCM is a useful solution.

Discussing how adding a DAG as input constraint changes things vs. not would be appreciated. If a fully-confounded SCM does not exist, then no DAG-compliant SCM exists etc.

**Limitations:**

The paper could be strengthened by adding a discussion on limitations, and addressing the questions posed above.

**Strengths And Weaknesses:**

**Strengths**
- Clear and clean presentation of the results and literature review. Figure 1 is appreciated.
- Important contributions to the hardness of a key class of problems.
- The "canonical" LP formulation of the problem is a useful feature to exploit instead of DP.

**Weaknesses**
- The motivation for some of the problems explored could be made clearer - see my questions below.

---

> ### Author Rebuttal · Authors · 2026-03-27
>
> Thank you for your helpful feedback.
>
> Regarding Question 1, please see our response to Reviewer YKHJ for a concrete example involving polynomials. We note that when polynomials are involved, we only establish  intractability - our main positive results apply to less general fragments.
>
> Regarding Question 2, it is indeed quite natural to assume that further restrictions – e.g., with respect to the underlying graph structure – are placed on the SCM that needs to be found. One can either account for this information by creating additional constraints (as we demonstrate in our response to Reviewer YKHJ), or by altering the problem setting in a way that now assumes the desired DAG to be given as an additional input. Very recently, the classical computational complexity of this problem version has been studied [1]; lifting these novel results to the parameterized realm poses an interesting challenge for future research. On a related note, a similar idea has also been used, e.g., in the setting of Bayesian Network Structure Learning (called the “superstructure” in theoretical studies). We will add a remark about this general direction in the concluding remarks of the final version.
>
> Reference
>
> [1] Markus Bläser, Julian Dörfler, Maciej Liśkiewicz, and Benito van der Zander. Probabilistic and Causal Satisfiability: Constraining the Model",  ICALP 2025

---

> > ### Author Rebuttal · Reviewer_jmob · 2026-04-02
> >
> > Thanks for your response!
> >
> > Question 1. This is my main reservation - what is the relevance and motivation of this work. This remains unanswered in my mind after your rebuttal, and I don't get it from re-reading the Intro section.
> >
> > > please see our response to Reviewer YKHJ for a concrete example involving polynomials.
> >
> > I may have missed something, but I see an example of linear constraints, not polynomial. And even then, my question is about *motivation* - why would one care in machine learning whether $\lambda.Pr(c' , b) \geq Pr(a')$? Or indeed higher powers like $\lambda.Pr(c' , b)^n \geq P(a')$?
> >
> > Question 2. This is not the main factor that is determining my score, and I appreciate the engagement. But could you elaborate on your comment below?
> > > One can either account for this information by creating additional constraints (as we demonstrate in our response to Reviewer YKHJ).
> >
> > How would a graphical constraint be incorporated as an additional linear/polynomial constraint?

---

> > > ### Author Response · Authors · 2026-04-07
> > >
> > > For Question 1, we believe the 4th and 5th paragraphs of our response to Reviewer YKHJ do provide such an example. In particular, the two paragraphs starting with "Now consider the following identification task: " show how a fundamental question - such as determining the impact of an intervention - can be expressed via the Satisfiability problem in the PCH using polynomials (for instance, the last constraint is a polynomial). Moreover, polynomials can be used to express, e.g., conditional probabilities in the PCH. Nevertheless, we do not intend to make a claim of strong practical relevance for the polynomial fragment of the PCH: our single result for that fragment merely rules out the possibility of lifting our other algorithms to the polynomial fragment, which has been introduced and studied in multiple previous works.
> > >
> > > For Question 2, we provide an exact construction in the 3rd paragraph of our response to Reviewer YKHJ, which we quote below:
> > >
> > > "Assume that we want to restrict our attention to SCMs that are compatible with, e.g., a DAG $G$ with vertices $W,X,Y,Z$ and directed edges $W \rightarrow X \rightarrow Z \rightarrow Y$, and $W \rightarrow Y$. One can encode this restriction by adding a constraint for each edge, i.e., for $X \rightarrow Z$ one adds the constraint $\sum_{x,z} (Pr([Z=z]X=x)-Pr(X=x))^2=0$ (which forces causal independence in the opposite direction)."

---

### Official Review · Reviewer_W4qt · 2026-03-13

**Soundness:** 4
**Presentation:** 4
**Significance:** 2
**Originality:** 4
**Overall Recommendation:** 5
**Confidence:** 3

**Summary:**

The paper considers the satisfiability problem for sets of inequalities whose terms may consist of observed, interventional or counterfactual probabilities for a given set of variables. A solution is given by an SCM that specifies all these probabilities so that all inequalities are satisfied. The paper establishes the existence of fixed-parameter tractable algorithms that decide the satisfiability of these inequalities across settings with varying restrictions based on the levels of Pearl's hierarchy and the permitted operations on the probability terms. It also shows that their results cannot be improved by either 1) dropping one of their fixed parameters or 2) extending their respective FPT results to a less restricted setting.

**Compliance With Llm Reviewing Policy:**

Affirmed.

**Final Justification:**

The rebuttal addressed my concerns and I improved my score to 5.

Even though I agree with other reviewers that the immediate practical implications of the presented results are unclear, I still think that the paper is quite strong due to the variety of new results that it presents and the non-trivial proof techniques that are being used.

In my opinion, the only reason to reject this paper would be if it is deemed out of scope for ICML, but I will let someone else make a final decision on this.

**Key Questions For Authors:**

1. Could you expand on why the satisfiability problem is useful in practice? In particular, I wonder if it makes sense to relate the Satisfiability problem to the causal discovery problem in settings where observational or interventional information is not precisely available but only constrained, e.g. in the spirit of [1]. Could there be any interesting setting in practice where the treewidth of the formula is indeed small?

[1] Guo, Z., & Dong, F. (2026). Linear Causal Discovery with Interventional Constraints. Machine Learning, 115(3), 35. https://doi.org/10.1007/s10994-026-06998-z

2. To gain a better intuition for the problem, it would be nice to see a natural example of a formula that is not satisfiable for a non-trivial reason (i.e. not something like $Pr(V = v) \geq 2$).

**Limitations:**

yes

**Strengths And Weaknesses:**

Strengths:
- The paper is mathematically very precise. All necessary concepts are rigorously defined. Proofs are well-written and correct (although I did not check all the proofs that are deferred to the appendix).
- The paper thoroughly discusses related work and significantly advances the state-of-the-art knowledge on the complexity of the considered satisfiability problem.

Weaknesses:
- Given that that the authors chose to submit to ICML instead of a TCS conference, the significance of the results for the ML community could be better addressed. The authors do not explain in which settings the satisfiability problem needs to be solved in practice.
- Even though the proposed algorithms improve our understanding of the hardness of the considered satisfiability problems, especially the FPT algorithm for $SAT_{counterfactual}^{lin}$ is completely infeasible to use in practice (apart from tiny problem instances)
- Instead of having an appendix that is nicely compatible with the main text, the paper simply refers to a longer version of itself in the supplementary material. I don't think this is the intended use of the supplement. However, the authors did make a good effort of providing comprehensive proof sketches for the omitted proofs.

Minor comments:
- I think it should be mentioned in the introduction that the paper effectively only considers discrete probability distributions
- I'm assuming the $\top$ symbol in the definition of $\mathcal{E}_\text{int}$ stands for an empty intervention set, but this might deserve a short explanation
- The complexity class W[1] (Theorem 4.3) is not defined but I don't think it should be assumed that the reader is familiar with it
- Since the definition of nice tree decompositions is non-standard and absolutely crucial for understanding the proof of one of the main results Theorem 4.1, it should really be in the main text

---

> ### Author Rebuttal · Authors · 2026-03-27
>
> Thank you for the detailed comments.
>
> Regarding the significance of the satisfiability problem for the ML community, this has been discussed in the related works that serve as a precursor to ours, but you are correct that we could have provided more information on this in our paper. We will do so in the final version.
>
> Regarding the appendix, when preparing our paper we verified that full-version appendices were allowed at ICML and have in fact been used in the past. We concede that this is a matter of taste; one advantage of a full-version appendix is that it more easily facilitates the inclusion of additional information beyond missing proof details (including, e.g., extra examples).
>
> We would also like to thank you for pointing us towards the recent work on linear causal discovery. To the best of our knowledge, the formalism chosen in [1] is very different from ours: there, the pairwise relation between variables is assumed to be represented as a two-dimensional matrix whose entries $(i,j)$  indicate whether intervening on variable $X_i$ has a positive or negative effect on variable $X_j$. If we understand correctly, such a positive effect could be translated into our setting via the constraint $Pr([X_i=x] X_j=x’) > Pr(X_j=x’)$. In this sense, PCH languages can indeed be used to specify desired interventional behavior as well as structural properties of the underlying graph (see our comment to Reviewer YKHJ).
>
> Regarding treewidth, we expect that the treewidth may be small in settings with rather local constraints in the following sense: The set of constraints consists of multiple smaller parts which each speak only about a few variables, and where only few variables link these distinct parts together to create interdependencies. While one could expect such instances to be reasonably common as interconnected smaller subproblems may be present in many different settings, we are not aware of any concrete empirical studies of treewidth in practically used causal models.
>
> Regarding Question 2, here is an example of a more complex unsatisfiable formula. Consider three random variables $A, B, C$ over a binary domain and the following constraints:
>
> - $Pr(A = 0 \lor B = 0) \le 0.5$
> - $Pr(C = 0 \lor B = 0) \ge 0.7$
> - $3\cdot Pr(C = 0 \land B = 1)  \le Pr(A = 0)$
>
> This is unsatisfiable as follows:
> The first inequality gives $Pr(B = 0) \le 0.5$, so by the second inequality we have $Pr(C = 0 \land B = 1) \ge 0.2$.  Using the last inequality this yields $Pr(A = 0) \ge 0.6$, contradicting the first inequality.
>
>
> We will incorporate all the suggestions in the minor comments into the final version - thank you for letting us know about them.
>
> Reference
>
> [1] Zhigao Guo and Feng Dong. Linear Causal Discovery with Interventional Constraints. Machine Learning Volume 115. 2026

---

> > ### Author Rebuttal · Reviewer_W4qt · 2026-04-02
> >
> > Thank you for addressing my questions. The only point that I'm still unsure about is the significance for the ML community. You state that this is discussed in previous papers and that you will add something to the final version. Could you summarize what you would add in a few sentences? Thanks a lot.

---

> > > ### Author Response · Authors · 2026-04-07
> > >
> > > In the final version of our paper, we will first elaborate on the significance of the satisfiability problem in PCH by clarifying that tasks such as causal identification as well as enforcing other properties of SCMs can be phrased as a question of satisfiability in PCH (building on the exposition in [1]). Secondly, we will emphasize that studying the complexity of different flavors of the satisfiability problem - in particular in a structure-aware manner - has led to insights that were exploited in practical solving in the past, for instance in the SAT- and CSP-community. Our work indicates that in applications where only the "lower" fragments of PCH arise, it is worth carrying out a structural analysis of the respective instances before feeding them to a general solving procedure, as we have reason to believe that on certain “nicely structured” instances more efficient solving procedures exist. We will make explicit that an analysis of the likeliness with which such nicely structured instances occur in practice has yet to be carried out, but to motivate such a study in the future, we will point again to results from other communities (e.g. SAT) showing that a significant amount of practically relevant instances are well-structured.
> > >
> > > [1] Benito van der Zander, Markus Bläser, and Maciej Liśkiewicz. The hardness of reasoning about probabilities and causality. IJCAI 2023.

---

### Official Review · Reviewer_podn · 2026-03-13

**Soundness:** 3
**Presentation:** 2
**Significance:** 1
**Originality:** 3
**Overall Recommendation:** 4
**Confidence:** 1

**Summary:**

This is a paper about the computational complexity of inference in Pearlian structural causal models. The authors consider the popular ladder of causation (observational/interventional/counterfactual inferences) and derive dedicated complexity results related to (causal, but also Boolean) satisfiability theory.

**Compliance With Llm Reviewing Policy:**

Affirmed.

**Final Justification:**

After reading the rebuttal, the other reviews and the corresponding discussions, I decided to raise my overall score to 4 (weak accept) while also confirming my low confidence. Now I better understand the potential impact of practical causal analyses and, even if it lack an empirical part, it seems the paper can be safely accepted.

**Key Questions For Authors:**

Is it possible to see an example of queries involving polynomials over the probabilities of practical interest for causality?
Is it possible to also take into consideration also nested counterfactuals?
Bareinboim recently considered intermediate layers of the ladder ("2.5"). This could be somehow connected to the analysis presented here.

**Limitations:**

-

**Strengths And Weaknesses:**

I should say that I am a specialist of causal inference, but my knowledge in computational complexity is very basic, and this review (as shown by my confidence assessment) should be intended only as an educated guess.

In terms of soundness, I cannot really evaluate the correctness of the proofs. I can only observe that the results are compatible with the complexity results I know about causal inference in SCM (see, e.g., Zaffalon et al., IJAR 2024).

The presentation is very hard to grasp, but my very low confidence might be the reason.

In terms of significance, I should say that I am not sure this kind of complexity-inspired approaches can eventually lead to practical algorithms to be used by people in causality. From this perspective, the lack of an empirical part and/or examples does not help. This is the main reason for my negative recommendation, which, I repeat, should be properly weighted by my low confidence.

---

> ### Author Rebuttal · Authors · 2026-03-27
>
> Thank you for the honest review. You are correct that the paper is of theoretical nature and does not include experiments. We note that ICML has historically also welcomed purely complexity-theoretical papers on the core topics of the community; like ours, these often provide a combination of intractability results (ruling out efficient algorithms) and algorithmic upper bounds (which may eventually lead to practical algorithms).
>
> Regarding Question 1, please see the response to Reviewer YKHJ for one such example. That being said, our results indicate that involving polynomials over the probabilities leads to intractability while the linear fragments are more amenable to algorithmic techniques.
>
> Regarding Question 2, nested counterfactuals allow for counterfactual statements within the premise or conclusion of a counterfactual. In particular, this class of languages contains all counterfactual statements which means that our hardness results for the counterfactual case carry over. We would not be surprised if the fixed-parameter tractability result (Theorem 5.3) obtained for the counterfactual fragment can also be lifted to the setting with nested counterfactuals, but verifying whether this is true would need to be carried out in future work.
>
> Finally, we would like to thank you for pointing us towards the recently introduced intermediate layers in the PCH [1] which we were previously not aware of. As far as we understand, the intermediate layers 2.25 and 2.5 are established by fixing interventional variable sets and value sets, respectively, thereby restricting the types of possible interventions. Note however, that our complexity results for causal and counterfactual fragments of the PCH coincide; that is, with respect to the parameters that we considered, the satisfiability variants for these languages admit the same complexity-theoretic lower and upper bounds. Consequently, these results also carry over to the intermediate layers 2.25 and 2.5. We will add a remark on this in our final version.
>
> Reference
>
> [1] Hongshuo Yang and Elias Bareinboim. A Hierarchy of Graphical Models for Counterfactual Inferences. NeurIPS 2025

---

> > ### Author Rebuttal · Reviewer_podn · 2026-04-02
> >
> > Thanks for the clear rebuttal. I better understand the potential of this contributions and I am considering to raise my recommendation (but I will better do that after a discussion with the other reviewers and the AC).

---

### Official Review · Reviewer_YKHJ · 2026-03-16

**Soundness:** 3
**Presentation:** 3
**Significance:** 3
**Originality:** 3
**Overall Recommendation:** 5
**Confidence:** 3

**Summary:**

The paper makes the following claims regarding $SAT^{\mathrm{lin}}_{\mathrm{prob}}$, the satisfiability problem for linear combinations of observational distributions -- whether or not there exists an SCM satisfying a given boolean formula of the form $\sum_i P(\mathbf Z_i) \leq 0$, for conjunctions of atoms $Z_i$:
1. $SAT^{\mathrm{lin}}_{\mathrm{prob}}$ is in $XP$, running in time $|I|^{f(k)}$ where $|I|$ denotes the input size and $f, k $ are a computable function and a specified numerical parameter.
2. It is fixed-parameter tractable w.r.t. the primal treewidth $tw$ plus domain size $d$.
3. It is impossible under some assumptions to improve the XP-tractability to fixed-parameter tractability (FPT), even for $SAT^{\mathrm{base}}_{\mathrm{prob}}$ (where only single probability terms are allowed, rather than a linear combination)
4. It is impossible under these assumptions to lift any of the tractability results to polynomial combinations of observational distributions or linear combinations of interventional distributions.

As a second set of contributions, the paper makes claims about $SAT^{\mathrm{lin}}_{\mathrm{counterfact}}$, linear combinations of counterfactual expressions.

5. It is fixed-parameter tractable w.r.t. number of variables in the formula $n$ plus domain size $d$.
6. It is impossible under some assumptions to drop any parameters while preserving FPT, even for $SAT^{\mathrm{base}}_{\mathrm{prob}}$.
7. It is impossible under these assumptions to lift the FPT to $SAT^{\mathrm{poly}}_{\mathrm{counterfact}}$.

**Compliance With Llm Reviewing Policy:**

Affirmed.

**Key Questions For Authors:**

1. Is the "expressivity matrix" for computation on SCMs complete? For example, does it capture expressions containing division, logarithms, or exponents? What is the reason for specifically choosing linear and polynomial combinations of probability expressions?
2. What are the impacts of solving causal satisfiability on other causal inference tasks, such as identification or estimation?

**Limitations:**

Yes.

**Strengths And Weaknesses:**

**Soundness.** The theoretical results in the paper appear to be correct.

**Presentation.** The paper is well-structured. Notation and contributions are clearly motivated, stated, and discussed.

**Significance.** The paper addresses the issue of tractability in evaluating observational, interventional, and counterfactual queries, and through its results in the linear setting (and impossibility result in polynomial settings), it may be a valuable stepping stone to more general causal satisfiability algorithms.

**Originality.** The work's core claims appear to be novel. In addition, the work uses a novel linear programming based technique to construct their proofs.

---

> ### Author Rebuttal · Authors · 2026-03-27
>
> Thank you for the encouraging comments.
>
> Regarding Question 1, the expressivity matrix as considered in our paper is the same as the one used in previous foundational papers. Some recent papers have considered an extension of the PCH to marginalization, which we mention as an interesting avenue for future work. Note that divisions can be expressed by moving terms to the other side of an equation, e.g. $Pr(X=x)/Pr(Y=y) = Pr(Z=z)$ can be expressed as $Pr(X=x)= Pr(Z=z) \cdot Pr(Y=y)$. Logarithms and exponents might be considered as further generalizations of the polynomial fragment (in particular, the polynomial fragment is simply the restriction of the exponential one to constant exponents). Nevertheless, our hardness results - and in particular Theorem 3.2 - show that even very simple instances remain intractable already within the polynomial fragment; this may be why neither we nor previous works have considered such generalizations of the expressivity matrix yet. More details on this are provided on lines 64-66 and lines 148-164 in the paper.
>
> Regarding Question 2, solving causal satisfiability can indeed be harnessed to perform other tasks in the setting of causality. For example, in the case of identification this is demonstrated in [1].
>
> For a concrete example, let us first note that certain graph properties can be expressed as constraints within our languages. Further, assume that we want to restrict our attention to SCMs that are compatible with, e.g., a DAG $G$ with vertices $W,X,Y,Z$ and directed edges $W \rightarrow X \rightarrow Z \rightarrow Y$, and $W \rightarrow Y$. One can encode this restriction by adding a constraint for each edge, i.e., for $X \rightarrow Z$ one adds the constraint $\sum_{x,z} (Pr([Z=z]X=x)-Pr(X=x))^2=0$ (which forces causal independence in the opposite direction).
>
> Now consider the following identification task: assume that we want to find the causal effect of the intervention $X=x$ on the outcome $Y=y$, that is, we want to compute $Pr([X=x]Y=y)$. It is well-known [2] that we can estimate this probability via front-door adjustment when we assume that $X$ and $Z$ are unconfounded and that every backdoor from $Z$ to $Y$ is blocked by $X$. Under these assumptions and for the graph considered above, we can compute $Pr([X=x]Y=y)$ by computing $\sum_z Pr(Z=z | X=x) \sum_{x'} Pr(Y=y | X=x', Z=z)Pr(X=x')$.
>
> Therefore, we can enforce the relation between $X$ and $Y$ to be amenable to front-door adjustment by adding $Pr([X=x]Y=y)=\sum_z Pr(Z=z | X=x) \sum_{x'} Pr(Y=y | X=x', Z=z)Pr(X=x')$ as an additional constraint. Further basic properties such as the fact that the joint probability distribution should factorize according to the graph’s structure (Markov Factorization) can also be specified – in the case of the graph $G$ above by adding the constraint $\sum_{w,x,z,y} (Pr(W=w, X=x, Z=z,Y=y) - Pr(U=u)Pr(X=x|U=u)Pr(Z=z|X=x)Pr(Y=y|Z=z,U=u))^2=0$.
>
>
> References:
>
> [1] Benito van der Zander, Markus Bläser, and Maciej Liśkiewicz. The hardness of reasoning
> about probabilities and causality. IJCAI 2023.
>
> [2] Judea Pearl. Causal diagrams for empirical research. Biometrika, 82(4):669–688, 1995.

---

> > ### Author Rebuttal · Reviewer_YKHJ · 2026-03-31
> >
> > Thanks for your detailed answers!

---

### Decision · Program_Chairs · 2026-04-30

**Decision:**

Accept (regular)

**Comment:**

All reviewers appreciate the theoretical underpinnings that the authors provide in this work, and especially praise the rigor in which this is done. They praise the clarity of the explanations the authors offered during the rebuttal, as these helped everyone to better understand the main contributions of the work. It is understood that this is a purely theoretical contribution, and that there is little point in asking for experimental validation. That said, there do remain questions regarding the practical relevance of the results. While it is certainly possible to do so, it is not clear why one would specify, eg. front-door admissibility constraints as a satisfiability problem, given that there exist polynomial-time algorithms to do so otherwise. Overall, based on the positive leaning of all reviewers, as well as appreciating results from theoretical computer science focused on machine learning problems myself, I recommend the paper to be accepted.